nature
human behaviour

ARTICLES

# The impacts of remote learning in secondary education during the pandemic in Brazil

Guilherme Lichand [1 ✉], Carlos Alberto Doria [1,2], Onicio Leal-Neto [1] and
João Paulo Cossi Fernandes[3]

The transition to remote learning in the context of coronavirus disease 2019 (COVID-19) might have led to dramatic setbacks in education. Taking advantage of the fact that São Paulo State featured in-person classes for most of the first school quarter of 2020 but not thereafter, we estimate the effects of remote learning in secondary education using a differences-in-differences strategy that contrasts variation in students' outcomes across different school quarters, before and during the pandemic. We also estimate intention-to-treat effects of reopening schools in the pandemic through a triple-differences strategy, contrasting changes in educational outcomes across municipalities and grades that resumed in-person classes or not over the last school quarter in 2020. We find that, under remote learning, dropout risk increased by 365% while test scores decreased by 0.32 s.d., as if students had only learned 27.5% of the in-person equivalent. Partially resuming in-person classes increased test scores by 20% relative to the control group.

n the decades before the coronavirus disease 2019 (COVID-19) pandemic, middle-income countries were making strides towards universal basic education. By 2019, enrolment rates for primary education had reached over 90% in Latin America and over 75% in Sub-Saharan Africa[1]. However, UNESCO and other international organizations described the global education outlook as a 'learning crisis'[2]. In middle-income countries such as Brazil, the setting of our study, even though most children are now in school, over half of 10-year-olds still cannot read age-appropriate texts[3], and 70% finish high school without minimum maths and language skills[4]. School closures in the context of the COVID-19 pandemic are expected not only to detrimentally affect such already fragile learning outcomes but also to upset recent progress in enrolment rates[5,6]. Despite widespread efforts to transition from in-person classes to remote learning[7–10], a multitude of factors combine to make the latter presumably much less effective in middle-income countries: limited internet access, lack of dedicated spaces to study at home and little support from parents, who often have not attended school for as long as their children have, above and beyond additional detrimental factors in the context of the pandemic, from the demand for child labour to violence against children in a context of psychological distress[11–14]. With over 1.6 billion children left without in-person classes for a prolonged period of time, international organizations estimate that at least 7 million additional students will no longer be in school when in-person classes return[15].

Quantifying learning losses due to remote learning within primary and secondary education is urgent, as governments need to make informed decisions when trading off the potential health risks of reopening schools in the pandemic[16] against its potential educational benefits. This remains to be the case even with high immunization coverage. In Brazil, while 49.4% of the population had received at least the first shot of the COVID-19 vaccine by July 2021, only approximately 25% of students had returned to in-person classes at that time[17]. Several papers have attempted to quantify learning losses from remote relative to in-person classes before the pandemic, but with important limitations when it comes to

generalizability. Most studies are based on high-income countries[18–20]. Out of those, some contrast online to in-person instruction within tertiary education[21–26], while those that focus on secondary schools restrict attention to charter schools, contrasting online to in-person instruction within a very selected sets of students[27–30]. In contrast, evidence for middle-income countries is thinner and mostly from experiments that use remote learning to expand educational access to rural and remote regions that had no access to education before[31–34], which is a very different counterfactual than in-person classes before the pandemic. In turn, the studies that try to estimate the extent of learning losses due to remote learning during the pandemic either rely on simulations and structural models[5,35,36] or suffer from comparability issues, contrasting different tests and student populations before and during the pandemic, and without parsing out other direct effects of COVID-19, above and beyond the transition to remote learning[37–47]. Even the few studies that rely on appropriate counterfactuals to study this question have to rest on strong assumptions, given the nature of the variation they use to identify causal effects. In particular, differences in the length of school recess across geographical units or that induced by previous epidemics[48,49] are only loosely related to the changes in instruction mode observed in the context of COVID-19. As such, the only credible evidence available for the impacts of remote learning on secondary schools during the pandemic is for high-income countries[50,51], leaving key questions unanswered, from the extent of its impacts on student dropouts (an issue that is relatively unimportant in high-income countries but critical for middle-income countries) to the extent of heterogeneity of those impacts with respect to student characteristics such as age, gender and race.

We contribute to this literature in several respects. First, we take advantage of a natural experiment in São Paulo State to provide estimates of the effects of remote learning under appropriate counterfactuals. In particular, our estimates allow for potential differences between remote and in-person examinations, and in the set of students taking them, by leveraging changes in instruction mode between Q1 and Q2–Q4 of the 2020 school year, while assessments

[1]Department of Economics, University of Zurich, Zurich, Switzerland. [2]Department of Economics, University of Brasília, Brasília, Brazil. [3]Inter-American Development Bank, Washington, DC, USA. ✉e-mail: guilherme.lichand@econ.uzh.ch

were kept fully remote across all school quarters in 2020. An additional natural experiment, linked to staggered school reopening in São Paulo State over the course of Q4 of 2020, further corroborates the evidence. Second, we provide evidence on the effects of remote learning in secondary education within middle-income countries, a context in which most school systems did not even conduct assessments for remote classes[52]. Third, we are able to parse out the effects of COVID-19 local disease activity to disentangle the effects of remote learning from other effects of the pandemic on educational outcomes (from health shocks to income losses to effects on students' mental health). Fourth, we document that the negative impacts of remote learning were larger for girls and for non-white students, and for schools in low-income neighbourhoods and those without previous experience with online academic activities.

## Approach and results

**Institutional background.** São Paulo State provides a unique opportunity to study this question for two reasons. First, the State Secretariat of Education, responsible for middle- and high-school students in the State, conducted quarterly standardized tests (*Avaliações de Aprendizagem em Processo* (AAPs)) throughout 2020, on the same scale as in the years before the pandemic. Such tests are not mandatory (although heavily promoted by the State, with a take-up rate no lower than 80% even during school closures). Supplementary Section C.2 shows that, while teachers might have tried to manipulate scorecard grades in 2020 (presumably in an attempt to prevent massive grade repetition), the same did not happen for standardized test scores.

Second, State schools transitioned to remote learning only at the very end of the first school quarter, when basically there were only examinations left to be taken before the start of the second quarter. This provides a natural experiment: in São Paulo, classes were in-person only in the first quarter (Q1) but remote thereafter (Q2–Q4), whereas examinations were remote across all school quarters. Remote examinations could be taken either online, through a zero-rating proprietary platform powered by the Secretariat of Education, or picked up and handed back in person at the school gate.

**Empirical strategy.** As such, we estimate the impacts of remote learning through a differences-in-differences strategy, contrasting the variation in the dropout risk and standardized test scores between Q1 and Q4 in 2020 relative to that in 2019, when all classes were in-person. Computing within-year variation not only absorbs teacher effects but also holds examination characteristics constant, as all examinations were remote in 2020 but in-person in 2019. We also present results of naive comparisons between Q4 of 2020 and Q4 of 2019, and of differences-in-differences analyses contrasting the variation in the dropout risk and standardized test scores between Q4 of 2019 and Q4 of 2020 relative to that between Q4 of 2018 and Q4 of 2019, both of which conflate the effects of other changes in 2020. All estimates are ordinary least-squares (OLS) regressions, absorbing grade fixed effects. We cluster standard errors at the school level, allowing random shocks to the outcomes of interest to be arbitrarily correlated over time within each school.

While there were other changes in standardized tests between 2019 and 2020 – in particular, the simplified curriculum recommended for Brazilian schools during the pandemic[53] was reflected in 2020 standardized tests[36] –, most importantly, such changes were not differential across school quarters: the AAP in Q1 of 2020 already reflected the simplified curriculum, benefiting from re-planning efforts that happened early on, as the state of the pandemic worsened in the country.

We refine our differences-in-differences strategy for the effects of remote learning on test scores by matching student characteristics across years, to parse out the effects of selection. We estimate a propensity score within each grade and quarter, based on student

characteristics (see Supplementary Section E), and control flexibly for it (with a cubic polynomial) to parse out selection. We also use these propensity scores to re-weight observations (by the inverse of their selection probability) to ensure that treatment effects on standardized test scores reflect those on the universe of students in each grade. We allow our estimates to vary non-parametrically with municipal-level per-capita COVID-19 cases and deaths over that period, to gauge the magnitude of treatment effects in the absence of other effects of disease activity on learning outcomes, finding that losses did not vary systematically with local disease activity.

We also estimate heterogeneous treatment effects of remote learning by student age, gender and race (coded in the Secretariat of Education's administrative data), by the average per-capita income of the neighbourhood where the school is located (according to the 2010 Demographic Census) and by whether the school offered online academic activities even before the pandemic (according to the 2019 Educational Census).

Last, we estimate the educational impacts of resuming in-person classes in the pandemic, taking advantage of a second natural experiment in São Paulo State. Over Q4 of 2020, some municipalities allowed in-person optional activities (psycho-social support and remedial activities for students lagging behind) to return for middle-school students and in-person classes to return for high-school students[16]. Since in-person classes only returned for the latter, we implement a triple-differences strategy, contrasting differences between middle- and high-school students within municipalities that authorized schools to reopen versus those within municipalities that did not, before and after school reopening. Because we do not have data on which schools actually reopened, we estimate intention-to-treat (ITT) effects based on municipalities' authorization decrees, through OLS regressions. Importantly, since neither COVID-19 cases nor deaths varied systematically across municipalities that reopened schools and those that did not[16], this strategy further validates our previous estimates of the causal educational impacts of remote learning in secondary education. Moreover, because the staggered return to in-person activities happened only in Q4, this additional natural experiment also allows us to test the hypotheses on whether learning losses from remote learning relative to in-person classes were only short-lived, that is, concentrated around the time of the transition but gradually fading out as teachers and students adapted. If this were the case, then we should see no positive educational impacts from resuming in-person classes at that point, nearly 8 months into remote learning.

**Data and definition of outcomes.** We have access to quarterly data on student attendance and maths and Portuguese scorecard grades, and standardized test scores for the universe of 6th to 12th graders in São Paulo State between 2018 and 2020. Restricting attention to 2019 and 2020, our data comprise 4,719,696 observations for middle-school students and 3,791,024 for high-school students. Because of selection into examinations each year, we have data on standardized test scores for 83.3% of observations.

Tracking student dropouts in the pandemic is challenging. Most education secretariats in Brazil automatically re-enroled students at the beginning of 2021[16]. As a leading example, while middle- and high-school dropouts in São Paulo State average 10% in a typical year, dropouts were officially 0% in 2021. Instead, we focus on dropout risk, which presumably looms regardless of official enrolment status. We define high dropout risk equal to 1 if a student had no maths and no Portuguese grades on record for that school quarter, and 0 otherwise. The rationale for defining dropout risk in this way is that abandoning school is often the outcome of a cumulative process of student disengagement with school activities[54–56]. Without reliable attendance data in the absence of in-person classes during the pandemic, keeping track of whether teachers at least imputed scorecard grades for the student is as good a measure

**Table 1 | Effects of remote learning on dropout risk and test scores**

| | Q4 2020 − Q4 2019 | (Q4 2020 − Q4 2019) − (Q4 2019 − Q4 2018) | (Q4 2020 − Q1 2020) − (Q4 2019 − Q1 2019) | | |
|---|---|---|---|---|---|
| | (1) | (2) | (3) | (4) | (5) |
| **Panel A**: High dropout risk | 0.0662 | 0.0691 | 0.0621 | 0.0621 | 0.0621 |
| | (0.0002) | (0.0002) | (0.0002) | (0.0002) | (0.0002) |
| | $P < 0.001$ | $P < 0.001$ | $P < 0.001$ | $P < 0.001$ | $P < 0.001$ |
| Mean for Q4 of 2019 | 0.017 | 0.017 | 0.017 | 0.017 | 0.017 |
| $N$ | 4,271,928 | 6,724,744 | 8,543,588 | | |
| **Panel B**: Standardized test scores | 0.652 | 0.523 | −0.314 | −0.301 | −0.319 |
| | (0.0001) | (0.0001) | (0.0001) | (0.0001) | (0.0001) |
| | $P < 0.001$ | $P < 0.001$ | $P < 0.001$ | $P < 0.001$ | $P < 0.001$ |
| In-person learning equivalent | 0.44 | 0.44 | 0.44 | 0.44 | 0.44 |
| $N$ | 3,688,042 | 6,367,375 | 7,097,042 | | |
| Grade fixed effects | Yes | Yes | Yes | Yes | Yes |
| Matching | No | No | No | Yes | Yes |
| Inverse probability weighting | No | No | No | No | Yes |

Notes: The table displays treatment effects of remote learning on educational outcomes. Column 1 compares Q4 of 2020 with Q4 of 2019. Column 2 compares the variation between Q4 of 2019 and Q4 of 2020 with that between Q4 of 2018 and Q4 of 2029. Columns 3–5 show estimated differences-in-differences comparing the variation in outcomes between Q1 and Q4 of 2020 with that between Q1 and Q4 of 2019. In panel A, the dependent variable is high dropout risk (=1 if the student had no maths or Portuguese grades on record for that school quarter, and 0 otherwise). In panel B, the dependent variable is scores from quarterly standardized tests (AAPs), averaging maths and Portuguese scores for that school quarter. All columns include grade fixed effects and an indicator variable equal to 1 for municipalities that authorized schools to reopen from September 2020 onwards, and 0 otherwise (allowing its effects to vary at Q4). In columns 4 and 5, we control for the propensity score of selection into examinations (see Supplementary Section E) with a third-degree polynomial. In column 5, we also re-weight observations by the inverse of their propensity score. All columns are OLS regressions, with standard errors clustered at the school level. $P$ values are computed from two-sided $t$-tests that each coefficient is equal to zero.

of disengagement with school activities as it gets. This and similar measures have also been used in the literature[57],[58–60] and by the State Education Secretary and philanthropic organizations that support quality education in Brazil (for example, to predict which schools are most likely to be affected by student dropouts[61]). In the Supplementary Information, we show that this proxy reliably predicts actual aggregate dropouts in the years before the pandemic (Section A1) and that students with missing scorecard grades in 2020 were much more likely not to have engaged in any academic activity in Q1 of 2021 (Section A2).

When it comes to standardized test scores, we average maths and Portuguese scores within each quarter (for Q4 of 2020, only overall standardized test scores are available). We retain in the sample only students with valid test scores in Q1 and Q4 of each school year. In the Supplementary Information, we estimate the impacts of remote learning separately for maths and Portuguese standardized test scores until Q3 of 2020 (Section E). Our main treatment variable indicates whether students were exposed to remote rather than in-person classes. As such, it equals 1 for Q2–Q4 of 2020, and 0 otherwise.

**Effects of remote learning.** Table 1 presents different estimates of the effects of remote learning on high dropout risk (in percentage points (p.p.), panel A) and standardized test scores (in standard deviations (s.d.), panel B). The first two columns present naive comparisons as a benchmark. Column 1 contrasts the last school quarters of 2019 and 2020, while column 2 contrasts the variation between the last school quarters of 2019 and 2020 versus its 2018–2019 counterpart. Next, columns 3–5 present our differences-in-differences estimates. Column 4 controls flexibly for the propensity score to parse out selection effects, while column 5 further re-weights observations by the inverse of their selection probability. Columns 3–5 also parse out the effects of school reopening over Q4 of 2020. The table suggests that remote learning might have had devastating effects on student dropouts, as measured by the dropout risk, which increased significantly during remote learning, by roughly

0.0621 (s.e. 0.0002), a 365% increase (significant at the 1% level, columns 3–5). Given the relationship between the proxy and actual dropouts (discussed in Supplementary Section A.1), this result is suggestive of student dropouts within secondary education in the State having increased from 10% to 35% during remote learning. In Supplementary Section A, we show that this finding is robust to correcting for measurement error based on administrative data, which allows us to compute false positives and false negatives both before and during the pandemic, and provide evidence that the proxy is indeed highly predictive of not attending any classes in Q1 of 2021, when in-person classes had been authorized to return by all municipalities of São Paulo State.

Regarding learning losses, naive comparisons across years would lead one to incorrectly conclude that test scores have actually increased during remote learning (columns 1 and 2), with effect size of 0.652 (s.e. 0.0001) or 0.523 (s.e. 0.0001), depending on the specification. These naive estimates reflect the fact that average grades were higher in 2020 than in 2019. There are two important factors that probably explain this pattern: first, a selection effect, linked to both higher student dropouts and differential selection into standardized tests during remote learning; second, differences in assessment mode between years. In particular, AAPs in 2020 covered a simplified curriculum, and students had much more time to take the examination in 2020 than in previous years (two days, relative to a couple of hours). See Supplementary Section C.1 for a detailed discussion of differences between in-person and remote examinations in the context of São Paulo State. The differences-in-differences strategy, in turn, uncovers dramatic losses of 0.32 s.d. (s.e. 0.0001), significant at the 1% level, a setback of 72.5% relative to the in-person learning equivalent. In panel B, columns 4 and 5 use propensity score matching to account for the potential selection of students into standardized tests based on characteristics, especially given the significant treatment effects of remote learning on dropout risk documented in panel A.

Reference [35] documents, in a different context, that differences between cohorts over time, in particular due to selection in the

context of COVID-19, can generate sizeable differences in measured learning outcomes throughout the pandemic. Not only does our empirical analysis compare how the same cohorts evolved over time, but also, our matching strategy in columns 4 and 5 ensures that the characteristics of students who took different examinations are balanced (Supplementary Table E.1). Results are very robust to the matching procedure. This is not because selection is unimportant; indeed, Supplementary Table E.2 shows that student characteristics matter for the differential take-up of standardized tests in 2020. Rather, this presumably reflects the fact that the nature of selection largely remains the same across Q1 and Q2–Q4 of 2020. As such, self-selection into the examinations has small to no impact on our main results. Supplementary Table E.7 additionally re-estimates the results in Table 1 using a balanced panel, by restricting attention to the students who took all the standardized tests in 2019 and 2020. Even within this highly selected sub-sample (given the results in panel A of Supplementary Table E.7), the effect size of remote learning on learning outcomes is still over 70% of that within the whole sample, corroborating that findings are not an artefact of selection in the context of our study.

The average school in São Paulo State remained closed for approximately 35 weeks throughout 2020. As such, our estimates imply that students lost approximately 0.009 s.d. of learning each week relative to in-person classes. This effect size is only slightly larger than that in ref. [50] but is at least fourfold that in ref. [41]. Estimates in ref. [50] imply learning losses of 0.3–0.4 s.d per year, which translate to 0.005–0.007 s.d per week. Estimates in ref. [41] suggest average learning losses of 3–5 percentiles over the course of a year. To make our estimates comparable, we ordered test scores in the baseline period (Q4 of 2019) and evaluated the estimates in Table 1 relative to the score of the median student in the previous year, scaling those losses linearly for one year so as to keep the time frame constant. On the basis of those estimates, remote learning would have led to a 22–25 percentile decrease relative to in-person classes, a dramatic effect size.

Figure 1a displays heterogeneous treatment effects on dropout risk, normalized with respect to its average for Q4 of 2019 within each grade, by grade (estimates based on column 5 of Table 1). This figure shows that, except for high-school seniors, dropout risk increased by at least 300% across all grades (panel A), suggestive of uniformly large impacts on dropout rates across most grades. The estimated increase in dropout risk is actually higher for high-school students but since their baseline dropout risk was much higher than those for middle-school students, the relative increase ends up being smaller in percentage terms. Figure 1b displays heterogeneous treatment effects on standardized test scores, normalized with respect to the average difference between Q1 and Q4 of 2019 within each grade. In panel B, learning losses are also homogeneously distributed, being 60% or higher across all grades, with no distinctive differences between middle- and high-school grades. Supplementary Section D showcases that the negative effects of remote learning are significantly concentrated in girls and non-white students, and in schools located in poorer neighbourhoods and those that did not offer online academic activities prior to the pandemic. In Supplementary Section E, we show that our results are robust to the use of an alternative baseline period and that learning losses were especially dramatic within maths: until the third school quarter of 2020, students had learned 40% of what they would have learned under in-person classes in Portuguese, but only 20% in maths classes.

Last, Supplementary Section F also shows that allowing treatment effects to vary with local disease activity barely changes our conclusions. To study heterogeneity, we first residualize variation in educational outcomes and in COVID-19 cases with respect to student and school characteristics (allowed to influence learning outcomes differentially in Q4). For COVID-19 cases, this approximates quasi-random variation in disease activity within the State, conditional on all the characteristics that we observe[62]. We then estimate

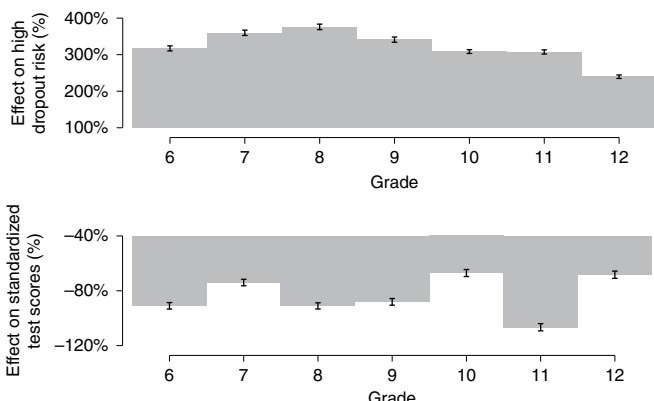

**Fig. 1 | Heterogeneous treatment effects of remote learning on dropout risk and standardized test scores by grade. a,b**, Effect sizes (bars) estimated through grade-specific OLS regressions using the differences-in-differences model, with 95% confidence intervals (error bars) based on standard errors clustered at the school level, where the dependent variable is high dropout risk (=1 if the student had no maths or Portuguese grades on record for that school quarter, and 0 otherwise, $N = 8,543,586$) (**a**) or scores from quarterly standardized tests (AAPs), averaging maths and Portuguese scores for that school quarter ($N = 7,097,042$) (**b**). All regressions follow the specification in column 5 of Table 1, only restricting observations to each grade. We normalize each effect size by its baseline mean, to express them as percentage changes. In **a**, the estimates are divided by the variation in the percentage of students with dropout risk = 1 between Q1 and Q4 of 2019 within each grade. In **b**, the estimates are divided by the variation in standardized test scores between Q1 and Q4 of 2019 within each grade. All columns include an indicator variable equal to 1 for municipalities that authorized schools to reopen from September 2020 onwards, and 0 otherwise (allowing its effects to vary at Q4), and a third-degree polynomial of propensity scores, and re-weight observations by the inverse of their propensity score.

heterogeneous treatment effects by flexibly regressing variation in these residualized educational outcomes across Q1 and Q4 on residualized municipal-level variation in per-capita COVID-19 cases over that period, with local polynomial regressions. We find that learning losses did not systematically increase with local disease activity over that period. In turn, while dropout risk does seem to have slightly increased with local disease activity (despite no statistically significant patterns), effect sizes were very large even at the low end of the distribution. We estimate that dropout risk increased by no less than 247% under remote learning conditions in the State. Also, differences at both ends of the distribution are not statistically different from each other. Supplementary Section F also shows that these results are robust to using COVID-19 deaths (much less prone to issues such as under-testing or under-reporting) instead of cases.

**Effects of resuming in-person classes.** In this section, we examine a second natural experiment. From October 2020 onwards, some municipalities allowed schools to reopen for in-person activities, following safe reopening protocols. In schools that actually reopened, from November 2020 onwards, high-school classes were allowed to resume in-person. In contrast, middle-school classes remained completely remote until the end of 2020. That motivates the additional, triple-differences strategy that we pursue in this section. Table 2 presents ITT estimates of the effects of resuming in-person activities on student attendance (in p.p., column 1), standardized test scores (in s.d., column 2) and dropout risk (in p.p., column 3), where in-person activities is an indicator variable equal to 1 if a municipality authorized schools to reopen, and 0 otherwise.

**Table 2 | ITT effects of in-person school activities on student attendance, dropout risk and standardized test scores**

| | (1) | (2) | (3) |
|---|---|---|---|
| | **Attendance** | **Standardized test scores** | **Dropout risk** |
| **Panel A**: Diff-in-diff: middle-school in-person activities | 0.010 | 0.001 | 0.001 |
| | (0.001) | (0.001) | (0.001) |
| | $P < 0.001$ | $P = 0.35$ | $P = 0.29$ |
| **Panel B**: Diff-in-diff: high-school in-person activities | 0.007 | 0.024 | 0.002 |
| | (0.001) | (0.0001) | (0.002) |
| | $P < 0.001$ | $P < 0.001$ | $P = 0.39$ |
| **Panel C**: Triple-differences in-person activities | −0.002 | 0.023 | 0.001 |
| | (0.002) | (0.001) | (0.001) |
| | $P = 0.04$ | $P = 0.001$ | $P = 0.31$ |
| Grade fixed effects | Yes | Yes | Yes |
| Matching | Yes | Yes | Yes |
| N | 3,701,482 | 2,624,943 | 3,701,482 |

Notes: The table displays ITT estimates of resuming in-person school activities on student attendance (column 1), standardized test scores (column 2) and high dropout risk (column 3). Quarterly data on attendance reflect online or in-person attendance and/or assignment completion (handed in online or in-person) over each quarter (in p.p.), averaged across maths and Portuguese classes; standardized test scores from quarterly standardized tests (AAPs), averaging maths and Portuguese scores for that school quarter; and high dropout risk = 1 if the student had no maths or Portuguese grades on record for that school quarter, and 0 otherwise. Panels A and B estimate treatment effects through differences-in-differences, contrasting the variation in outcomes between Q1 and Q4 of 2020 within municipalities that authorized schools to reopen versus those that did not. Panel A restricts attention to middle-school students, and panel B to high-school students. Panel C estimates treatment effects through a triple-differences estimator, which contrasts the differences-in-differences estimates for middle- and high-school students (for whom in-person classes could resume within municipalities that authorized schools to reopen in Q4 of 2020). Column 2 controls for a third-degree polynomial of propensity scores, and re-weights observations by the inverse of their propensity score. All columns are OLS regressions, with standard errors clustered at the municipality level. $P$ values are computed from two-sided $t$-tests that each coefficient is equal to zero.

The table contrasts municipalities that authorized schools to reopen versus those that did not, before and after school reopening, restricting attention to middle-school students in panel A and to high-school students in panel B. Panel C further contrasts those differences, as in-person classes only returned for the latter.

The table shows that school reopening increased student attendance by a similar extent for middle- and high-school students, by 1 p.p. (s.e. 0.001) for the former and 0.7 p.p. (s.e. 0.001) for the latter. Effect sizes are small (column 1), since attendance in 2020 captured a combination of in-person attendance, online attendance and assignment completion (through an app or handed in at the school gate), averaging over 90% across all school quarters. In face of that, the fact that school reopening increased attendance at all is testament that reopening indeed allowed some students to return to in-person activities. Most importantly, we find positive treatment effects on learning, fully driven by high-school students. In municipalities that authorized high-school classes to return from November 2020 onwards, test scores increased on average by 0.023 s.d. (s.e. 0.001, significant at the 1% level; panel C, column 2), a 20% increase relative to municipalities that did not. Incidentally, the absence of treatment effects on middle-school test scores rules out differential trends in educational outcomes across municipalities that authorized in-person classes to resume and those that did not. In municipalities that authorized schools to reopen for in-person

academic activities in 2020, the average school could have done so for at most 5 weeks. Thus, resuming in-person classes contributed to an increase in test scores of at least 0.005 s.d. per week. While this effect size is lower than that estimated in the previous section, in truth it is remarkably high (actually just the same as in ref. [50]), especially once accounting for the fact that it is based on ITT estimates, as we do not have data on which schools actually reopened (and for how long) in the municipalities that issued authorization decrees.

In Supplementary Section E, we show that these results are robust to using the number of weeks with in-person classes as the treatment variable and to matching observations based on municipal-level baseline characteristics.

## Discussion

In this paper, we provide evidence that remote learning in secondary education might have not only imposed severe learning losses for students who remained in school by the time in-person classes resumed (a nearly 75% setback relative to in-person classes) but also dramatically increased dropout risk, threatening to reverse decades of efforts to ensure nearly universal basic education in such countries. Since usual dropout rates within secondary education in São Paulo State are close to 10%, and since the correlation between dropout risk and actual dropouts before the pandemic was approximately 0.7, our results are suggestive that dropout rates among middle- and high-school students in the State could have reached 35% in 2021. A meta-analysis of learning losses during the pandemic[63] documents that our results lie at the high end of the estimated impacts on learning losses and student dropouts during the pandemic. This is not only because ours is one of the few papers to focus on secondary education but also because schools were closed in Brazil for longer than in almost any other country.

The lion's share of the impacts on test scores that we estimate took place over the first quarter of remote classes. This might lead to the concern that the learning losses we document were not the result of remote learning itself but rather of the emergency transition from in-person to remote. If that were the case, then we should expect that, by the end of the school year, when students and teachers had had time to adapt to the new instruction mode, the effectiveness of remote learning would converge to that of in-person classes. This hypothesis is, however, inconsistent with two key findings of the paper. First, dropout risk surged after Q3, inconsistent with the idea of a transitory shock. Second, the return of in-person activities in November, already 8 months into remote learning, significantly increased high-school students' test scores. As discussed, the effect size is very large, especially given that schools were open only for a short period of time and that we can only estimate ITT effects. As such, our interpretation is that the early onset of learning losses is rather linked to non-linear treatment effects: the magnitude of learning losses was so large at Q2 that it simply was not possible that test scores would keep deteriorating at the same pace thereafter.

A limitation of our analyses is that we will not know the extent of actual dropouts until later in 2022 (or even until 2023), when school systems will no longer re-enrol students automatically (which happened exceptionally in the context of the pandemic). Alternatively, we have relied on missing scorecard grades, a proxy used by the Education Secretariat and its philanthropic partners to identify students at high dropout risk. This proxy is also in line with the literature that attempts to predict student dropouts by measuring their lack of engagement with school activities[64]. Supplementary Section A documents that this proxy is, in fact, predictive of actual dropouts. In 2019, students with missing scorecard grades were approximately seven times more likely not to be enroled in the following year than other students. Nevertheless, the proxy is not flawless. Relying on it generates both false positives and false negatives even during in-person classes. Moreover, during the pandemic, missing scorecard grades might rather reflect transitory shocks that prevent those

students from handing in homework or taking examinations but not necessarily imply that they will drop out of school. Having said that, using administrative data for the first school quarter in 2021, when in-person classes had been authorized to return by all municipalities of São Paulo State, Supplementary Section A shows that students with missing scorecard grades in the previous year were almost nine times more likely not to attend a single class across all subjects in the following year, relative to other students, thus corroborating the validity of our proxy even during the pandemic. It also estimates the effects of remote learning while correcting directly for classification error, leveraging administrative data that allow the computation of false-positive and false-negative rates both before and during the pandemic. In any case, using this proxy introduces an additional layer of uncertainty into the estimates and makes it difficult to compare our results directly with other estimates in the literature based on student dropouts measured from actual re-enrolment decisions. Most importantly, as emphasized in ref. [35], the long-term prospects of these students might still be altered by targeted public policies, from remedial education to cash transfers to active searching for out-of-school children and adolescents.

São Paulo State's response to the pandemic was typical. It closed schools from 16 March 2020 and did not reopen them until September that year. Remote-learning strategies were rolled out from April onwards, heavily based on broadcasting content on open television. The State's educational response to the pandemic was rated close to the median quality for the country[65]. As such, we expect these findings to generalize to other middle-income country settings. Besides its impacts on learning outcomes, remote learning is also expected to affect a multiplicity of other child development indicators, including their psychological well-being[66,67]. Other recent evidence for Brazil also indicates that learning losses through 2020 were dramatic[68]. Nevertheless, ref. [68] focused only on test scores and did not decompose learning losses into those caused by remote learning and those caused by other changes in 2020, including other direct effects of COVID-19 such as its economic and health consequences over the course of the pandemic. These other effects are presumably large. Moreover, while the impacts of remote learning on learning losses did not vary systematically with local disease activity, recent studies show that students might present difficulties in concentration, insomnia or neurological disorders in the aftermath of COVID-19 infections, even 60 days after diagnosis[69–71].

Without rigorous evidence to quantify the contribution of remote learning to those educational impacts, decisions about what to do about in-person classes in the pandemic have been largely influenced by the potential health costs of reopening schools, without weighing those against the educational costs of keeping them closed[72,73]. In effect, other economic activities for which it is easier to quantify losses from closures, such as shopping centres, bars and restaurants, have reopened systematically earlier (and for longer) than schools in middle-income countries[74].

Our results for the effects of resuming in-person classes also contribute to this important debate. While it has been questioned whether in-person classes would contribute to learning at all in such a complex scenario, we show that they do. Average test scores deteriorated by nearly three-quarters in municipalities that did not reopen schools but by only two-thirds in those that did, even if only partially and for a rather short period of time. While reopening schools only for optional in-person activities was not sufficient to prevent learning losses (as we find no improvements in learning outcomes of middle-school students, relative to the control group), in-person classes were. However, neither measure was sufficient to mitigate the dramatic effects of school closures on dropout risk.

As such, the public debate should move on from whether schools should be open to how to reopen them safely[75]: whether to prioritize school staff in the vaccination schedule, what is the appropriate but feasible personal protection equipment in place, what safe capacity

limits should be, what changes in school infrastructure are required (for example, to ensure appropriate ventilation) and how to adapt public transportation to mitigate contagion risks on the way to school. These are the very same discussions that have been part of the public debate about shopping centres, bars and restaurants since the onset of the pandemic.

Outside the context of the pandemic, many countries (from Brazil to the United States) have shown recent enthusiasm for online classes in primary and secondary education, in particular when it comes to the home-schooling debate[76,77]. The evidence base for the impacts of remote learning still lacks consensus, with mixed causal evidence for its impacts on learning outcomes relative to in-person classes[27–31,33], on the one hand, and a more positive outlook for its effects on the set of individuals who pursue further education[22], on the other. Related studies[32–34] that evaluated interventions connecting top teachers in the country to students with the help of technology do not reflect how remote learning is typically implemented, and estimate their effects under a counterfactual of no access to education that is ultimately very different from that of in-person classes. In effect, they find sizeable learning gains from such interventions, in sharp contrast to the dramatic learning losses that we document.

Although we do not have data on private schools, which serve only 20% of primary and secondary students in Brazil but tend to be better equipped and to serve wealthier families relative to public schools, our finding that even schools located in relatively wealthier neighbourhoods and better equipped to offer online academic activities suffered enormous learning losses under remote learning suggests that our conclusions might extend to a broader student population. However, further research is needed to better understand the heterogeneity in the impacts of remote learning, especially by parental education, which we do not observe in our data.

## Methods

**Ethics approval.** Approval for this study was obtained from the Institutional Review Board of the Department of Economics at the University of Zurich (2020-079).

**Participants.** We have access to educational outcomes of all 6th to 12th graders in public schools directly administered by the São Paulo State Secretariat of Education over the period from 2018 to 2020.

**Data collection.** Administrative data were shared by the São Paulo State Education Secretariat. Consent for data collection was obtained by the Education Secretary at the moment of student enrolment. No compensation was paid for obtaining data. We have access to quarterly data on student attendance in math and Portuguese classes, math and Portuguese scorecard test scores and overall standardized test scores for the universe of students between 6th and 12th grade. All statistical analyses were performed within the Secretariat's secure cloud infrastructure. Only summary statistics and regression results were directly accessible by the researchers, and no data with personal identifiers could be removed from the server. The Secretariat of Education also shared data on which municipalities in the State had issued decrees authorizing schools to resume in-person high-school classes from November 2020 onwards. In our main sample for the years 2019 and 2020, we have a total of 8,543,858 data points for approximately 2.2 million students. Of these, 50% are male, and their average age is 14.68 years.

**Measures.** We define high dropout risk equal to 1 if a student had no math and no Portuguese grades on record in that school quarter, and 0 otherwise. For Q4 of 2020, only overall standardized test scores are available. For all previous quarters, we average across math and Portuguese standardized test scores. We use attendance in the analysis of the effects of school reopening in the pandemic. This metric combines online and in-person attendance, and online or offline assignment completion (handing in homework through the app, or in-person at the school gate).

**Analysis method.** We estimate the impacts of remote learning through a differences-in-differences strategy, contrasting the variation in the dropout risk and standardized test scores between Q1 and Q4 of 2020 relative to that in 2019. We also present results of naive comparisons between Q4 of 2020 and Q4 of 2019, and of differences-in-differences analyses contrasting the variation in the dropout risk and standardized test scores between Q4 of 2019 and Q4 of 2020 relative to its 2018–2019 counterpart, both of which conflate the effects of other changes in 2020, in particular, changes in standardized tests from in-person to remote. We refine

our estimates of treatment effects on test scores by matching observations based on their propensity score, the predicted probability of taking the examination within each quarter and grade, based on student and school characteristics. We also re-weight observations by the inverse of their propensity score, to obtain estimates representative for the universe of students within each grade. All analyses absorb grade fixed effects. We cluster standard errors at the school level, allowing random shocks to the outcomes of interest to be arbitrarily correlated with schools.

We also estimate heterogeneous treatment effects by municipal-level per-capita COVID-19 cases and deaths over that period through non-parametric methods, after residualizing the variation with respect to student and school characteristics (allowed to influence learning outcomes differentially in Q1 and Q4), to parse out other effects of disease activity on learning outcomes.

Last, we estimate ITT effects of resuming in-person classes on educational outcomes also through a differences-in-differences strategy, but in this case contrasting municipalities which authorized schools to reopen for in-person activities versus those that did not, before and after in-person classes resumed for high-school students. We also undertake a triple-differences analysis, in which we contrast differences between middle- and high-school students within municipalities that allowed schools to reopen versus those within municipalities that did not, before and after school reopening. We cluster standard errors at the municipality level in these analyses. We rely on the central limit theorem to perform asymptotic inference and cluster standard errors at the school level. All hypothesis tests in this paper are two tailed.

**Reporting summary.** Further information on research design is available in the Nature Research Reporting Summary linked to this article.

## Data availability
The dataset that supports the findings of this study cannot be made available as it contains student personal identifiers (*Código do aluno*, which uniquely identifies each student to the Education Secretariat). As such, the data are not accessible outside of the cloud infrastructure of the São Paulo State Secretariat of Education. To request access to the data, interested researchers must contact SEDUC-SP.

## Code availability
Syntax for the central claims of the paper can be found at https://github.com/Carlosalbertobdc/school-reopening.

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

## Acknowledgements

This study was funded by the Inter-American Development Bank (IADB). We are grateful to the IADB and the São Paulo State Education Secretariat for celebrating the partnership that made it possible to access administrative data on student-level educational outcomes. We also acknowledge the São Paulo State Education Secretariat's staff members who contributed to this study, especially G. Corte, V. Georges and A. Mesquita Moreira Junior. We thank J. Christen for his help organizing the references. The content is solely the responsibility of the authors. This research was funded by the Inter-American Development Bank (IADB) as part of a partnership between IADB and the São Paulo State Education Secretariat.

## Author contributions

G.L., C.A.D., O.L.N. and J.P.C.F. take responsibility for the integrity of the data and the accuracy of the data analysis. G.L. and O.L.N. decided to publish the paper. C.A.D. was responsible for analysing the data. G.L., C.A.D. and O.L.N. drafted the manuscript. G.L. and C.A.D. contributed to the statistical analysis. C.A.D. led data management. G.L., C.A.D., O.L.N. and J.P.C.F. critically revised the manuscript. All authors had responsibility for the decision to submit for publication.

## Competing interests

G.L. and O.L.N. received fees from the Inter-American Development Bank (IADB) for the design of this study. J.P.C.F. is an IADB staff member. IADB had no role in the study design, data collection or analysis, decision to publish or preparation of the manuscript. C.A.B. declares no competing interests. Correspondence and requests for materials should be addressed to G.L.

## Additional information

**Correspondence and requests for materials** should be addressed to Guilherme Lichand.

# nature research

|   |   |
|---|---|

# Reporting Summary

Nature Research wishes to improve the reproducibility of the work that we publish. This form provides structure for consistency and transparency in reporting. For further information on Nature Research policies, see our Editorial Policies and the Editorial Policy Checklist.

## Statistics

For all statistical analyses, confirm that the following items are present in the figure legend, table legend, main text, or Methods section.

| n/a | Confirmed |   |
|---|---|---|
| ☐ | ☒ | The exact sample size (*n*) for each experimental group/condition, given as a discrete number and unit of measurement |
| ☐ | ☒ | A statement on whether measurements were taken from distinct samples or whether the same sample was measured repeatedly |
| ☐ | ☒ | The statistical test(s) used AND whether they are one- or two-sided<br>*Only common tests should be described solely by name; describe more complex techniques in the Methods section.* |
| ☐ | ☒ | A description of all covariates tested |
| ☐ | ☒ | A description of any assumptions or corrections, such as tests of normality and adjustment for multiple comparisons |
| ☐ | ☒ | A full description of the statistical parameters including central tendency (e.g. means) or other basic estimates (e.g. regression coefficient) AND variation (e.g. standard deviation) or associated estimates of uncertainty (e.g. confidence intervals) |
| ☐ | ☒ | For null hypothesis testing, the test statistic (e.g. *F*, *t*, *r*) with confidence intervals, effect sizes, degrees of freedom and *P* value noted<br>*Give P values as exact values whenever suitable.* |
| ☒ | ☐ | For Bayesian analysis, information on the choice of priors and Markov chain Monte Carlo settings |
| ☒ | ☐ | For hierarchical and complex designs, identification of the appropriate level for tests and full reporting of outcomes |
| ☐ | ☒ | Estimates of effect sizes (e.g. Cohen's *d*, Pearson's *r*), indicating how they were calculated |

*Our web collection on statistics for biologists contains articles on many of the points above.*

## Software and code

Policy information about availability of computer code

| Data collection | We used an R studio server (version 3.4.4) to connect and extract sensitive data for this project in a secure cloud-based environment. Codes are available at https://github.com/Carlosalbertobdc/school-reopening. |
|---|---|
| Data analysis | We subsequently use the same R server (version 3.4.4) to conduct the data analysis. Codes are also available at https://github.com/Carlosalbertobdc/school-reopening. |

For manuscripts utilizing custom algorithms or software that are central to the research but not yet described in published literature, software must be made available to editors and reviewers. We strongly encourage code deposition in a community repository (e.g. GitHub). See the Nature Research guidelines for submitting code & software for further information.

## Data

Policy information about availability of data

All manuscripts must include a data availability statement. This statement should provide the following information, where applicable:

- Accession codes, unique identifiers, or web links for publicly available datasets
- A list of figures that have associated raw data
- A description of any restrictions on data availability

The data that support the findings of this study was made available by the São Paulo Education Secretariat (SEDUC-SP). Restrictions apply to the availability of these data due to students' personal identifiers. Data access requires an agreement with SEDUC-SP.

# Field-specific reporting

Please select the one below that is the best fit for your research. If you are not sure, read the appropriate sections before making your selection.

☐ Life sciences  ☒ Behavioural & social sciences  ☐ Ecological, evolutionary & environmental sciences

For a reference copy of the document with all sections, see nature.com/documents/nr-reporting-summary-flat.pdf

# Behavioural & social sciences study design

All studies must disclose on these points even when the disclosure is negative.

| | |
|---|---|
| Study description | In this study, we provide a quasi-experimental quantitative assessment of the effects of school closures and reopening on students' standardized test scores and dropout risk. |
| Research sample | The study sample includes all students in secondary education in São Paulo State schools between 2018, 2019, and 2020. |
| Sampling strategy | We use data on the universe of State public school students. |
| Data collection | We only use administrative data that the São Paulo State Education Secretariat made available via a remote server. |
| Timing | We use administrative data collected from 2018 to 2020. |
| Data exclusions | We exclude observations with unavailable data on test scores or socioeconomic characteristics. This amounts to 8% of the full sample (1,135,526 observations). |
| Non-participation | We use data on the universe of State public school students. |
| Randomization | Students' were not randomly allocated into treatment and control groups. |

# Reporting for specific materials, systems and methods

We require information from authors about some types of materials, experimental systems and methods used in many studies. Here, indicate whether each material, system or method listed is relevant to your study. If you are not sure if a list item applies to your research, read the appropriate section before selecting a response.

## Materials & experimental systems

| n/a | Involved in the study |
|---|---|
| ☒ | Antibodies |
| ☒ | Eukaryotic cell lines |
| ☒ | Palaeontology and archaeology |
| ☒ | Animals and other organisms |
| ☒ | Human research participants |
| ☒ | Clinical data |
| ☒ | Dual use research of concern |

## Methods

| n/a | Involved in the study |
|---|---|
| ☒ | ChIP-seq |
| ☒ | Flow cytometry |
| ☒ | MRI-based neuroimaging |

