## [Peer Review File · Nature Human Behaviour]

Peer Review Information

Journal: Nature Human Behaviour

Manuscript Title: The Impacts of Remote Learning in Secondary Education during the Pandemic in Brazil

Corresponding author name(s): Guilherme Lichand

Reviewer Comments & Decisions:

Decision Letter, initial version:

26th July 2021

Dear Dr Lichand,

Thank you once again for your manuscript, entitled "The Impacts of Remote Learning in Secondary Education: Evidence from Brazil during the Pandemic", and for your patience during the peer review process.

Your Article has now been evaluated by 3 referees. You will see from their comments copied below that, although they find your work of potential interest, they have raised quite substantial (and in some cases, fundamental) methodological and conceptual issues. In light of these comments, we cannot accept the manuscript for publication, but would be interested in considering a revised version if you are willing and able to fully address reviewer and editorial concerns.

We hope you will find the referees' comments useful as you decide how to proceed. If you wish to submit a substantially revised manuscript, please bear in mind that we will be reluctant to approach the referees again in the absence of major revisions. We are committed to providing a fair and constructive peer-review process. Do not hesitate to contact us if there are specific requests from the reviewers that you believe are technically impossible or unlikely to yield a meaningful outcome.

If you wish to submit a suitably revised manuscript we would hope to receive it *within 8 weeks*. We understand that the COVID-19 pandemic is causing significant disruptions which may prevent you from carrying out the additional work required for resubmission of your manuscript within this timeframe. If you are unable to submit your revised manuscript within 6 months, please let us know. We will be happy to extend the submission date to enable you to complete your work on the revision.

- Include a "Response to the editors and reviewers" document detailing, point-by-point, how you addressed each editor and referee comment. If no action was taken to address a point, you must provide a compelling argument. This response will be used by the editors to evaluate your revision and sent back to the reviewers along with the revised manuscript.
- Highlight all changes made to your manuscript or provide us with a version that tracks changes.

[REDACTED]

Thank you for the opportunity to review your work. Please do not hesitate to contact me if you have any questions or would like to discuss the required revisions further.

Sincerely,

Arunas Radzvilavicius, PhD
Editor
Nature Human Behaviour

Reviewer expertise:

Reviewer #1: sociology of education

Reviewer #2: economics of education, school dropout

Reviewer #3: education policy, education in developing countries

REVIEWER COMMENTS:

Reviewer #1:

Remarks to the Author:

Following the worldwide school closures starting in March 2020, there has been an outpouring of studies collecting data on the time use and academic outcomes of children and youth who are kept at home. Many of these have been from Europe and the US, while data from other parts of the world have been lacking. This study presents evidence from Brazil (São Paulo), which offers an important case given the long duration of school closures and the developing country context.

The study is well-designed, using a differences-in-differences design that compares test scores before and after remote learning to corresponding test score growth during the same period in the previous year. It also implements a triple-difference design comparing differences between middle- and high-school students in municipalities where the latter group was allowed to return to school. The study finds large increases in incomplete grades (which they call "dropout risk") and drops in test scores, as well as improvements in test scores among high-school students who were allowed to return.

The study appears to confirm evidence from previous work. The design is close to that of Engzell et al (2021), and better than most other studies. This study is also valuable for estimating treatment effects across the age distribution, across population groups, and by local disease intensity. The finding that disease intensity is mostly unrelated to student outcomes is important since it counters a key objection to studies of learning loss in the wake of COVID: that they reflect wider impacts of the pandemic and not that of school closures as such. The added analysis of the effect of school reopening is well executed and does not have a parallel in existing literature.

My main concern is that the study is not appropriately contextualized with respect to existing studies of COVID-induced learning loss. There are by now dozens of studies on this subject (see bibliography below). Some of them are briefly discussed in the paper, but not until the concluding discussion. The motivation of this paper with respect to other works needs to be made clear already in the introduction. Specifically, the abstract and introduction overclaim when they say that "no study has rigorously documented the educational impacts of remote learning relative to in-person classes within primary and secondary education" and that "the evidence base for the impacts of remote learning is thin".

To my mind, the main contributions of this paper is that it a) uses sound data and methods to expand the evidence on COVID-induced learning loss, b) extends previous evidence centered on Europe and the US to a Latin American country, c) studies population heterogeneity along several dimensions, and d) studies the effect of school re-opening and not just closure.

In addition, I have several smaller comments on the analysis and discussion.

More information on the tests needs to be included. How were these designed and for how long did the students sit them? Is there any sense of the reliability? Did the remote testing regime offer any opportunities for cheating? Were the tests taken at Q1 and Q4 identical in design? If not, how was a difference score calculated and what is its interpretation?

To better allow comparison with existing work, effect sizes should be discussed in light of the exact length of school closures. What is the expected loss per week of school closure, and what is the expected gain per week of opening up? How does this compare to the estimates reported by Engzell et al (2021) for a European country or Kuhfeld et al for the US? (<https://www.nwea.org/content/uploads/2020/11/Collaborative-brief-Learning-during-COVID-19.NOV.2020.pdf>)

Test scores are markedly higher in 2020 compared to 2019. There are several potential reasons for this: sample selection, mode effects on the test (online/take-home vs in person), and the simplified curriculum during the pandemic. These points appear scattered throughout but need to be brought together at some point in the text. Do the authors view any of them as a more likely explanation?

The main results report difference-in-difference estimates with 2019 as a single comparison year. I assume that the results do not differ much if 2018 is used as a comparison instead, but this information could be included to gauge the robustness. What is the estimate of (Q4 2020-Q1 2020)-(Q4 2018-Q1 2018)?

The introduction and discussion mention that in municipalities in which high school students were allowed back, middle school students also partly returned. This information is not mentioned in the subsection "Effects of Resuming In-person Classes", so a reader who skips to that section will be puzzled about why attendance increased for both groups in Table 2.

The authors use incomplete grades as a proxy for dropout risk, arguing that enrollment was kept artificially high by school administrators. This seems fine. But, incomplete grades do not appear to be a very good predictor of dropout (Figure A.1). Please report the correlation in this figure. And as a suggestion, why not call the outcome incomplete grades instead?

Figure B.1 suggests that most of the shortfall in test score growth occurred between Q1 and Q2 when classes were supposedly still in person, and rates of learning were similar thereafter. Does this affect conclusions about the efficacy of remote learning? Should we conclude that educators and families struggled initially but eventually adapted with time?

Figure C.1-C.2. It seems unconventional to use a continuous density function to represent a discrete outcome. These graphs could be made clearer if a histogram was used.

In the analysis of heterogeneous treatment effects by student demographics shown in Appendix D, no coefficients or SEs are reported. There is little way to gauge the magnitude of these differences and their significance. Adding these numbers would be helpful.

Table E.1 reveals that non-white, poorer and low-performing students are underrepresented in 2020. The authors use propensity score reweighting to address this but confounding on unobservables might remain. However, the fact that adjustment for observed confounders does not alter estimates much between column (3) and (4)-(5) in Table 1 can be marshalled to claim that residual confounding is also likely to be small.

A key contribution of this paper is to study heterogeneity by local COVID incidence. The authors write that "risk increased with local disease activity" but Figure F.1 largely looks like the absence of a meaningful association to me. However, the detected case load depends on testing capacity. If learning loss is larger in poorer communities and testing capacity correlates negatively with poverty, null effects might be spurious. Is this a worry?

In referring to Supplementary Materials, please state the specific Figure/Table you are referring to.

Below is a partial list of relevant work. The authors do not need to cite all this, but they do need to revise the abstract, introduction and discussion to reflect the fact that they are not alone in studying the effect of school closures and remote learning during the pandemic.

- * Ahn, Kunwon; Lee, Jun Yeong; and Winters, John V., "Employment Opportunities and High School Completion during the COVID-19 Recession" IZA Discussion Paper
- * Boruchowicz, Cynthia et al "Time Use of Youth during a Pandemic: Evidence from Mexico"
https://www.researchgate.net/publication/350122372_Time_Use_of_Youth_during_a_Pandemic_Evidence_from_Mexico
- * Curriculum Associates, "Understanding Student Needs: Early Results from Fall Assessments"
<https://www.curriculumassociates.com/-/media/mainsite/files/i-ready/iready-diagnostic-results-understanding-student-needs-paper-2020.pdf>
- * Domingue, Hough, Lang, Yeatman, "Changing Patterns of Growth in Oral Reading Fluency During the COVID-19 Pandemic"
<https://edpolicyinca.org/publications/changing-patterns-growth-oral-reading-fluency-during-covid-19-pandemic>
- * GL Assessment, "Impact of Covid-19 on attainment – initial analysis "
<https://www.gl-assessment.co.uk/news-hub/research-reports/impact-of-covid-19-on-attainment-initial-analysis/>
- * Gore, Jennifer, Leanne Fray, Andrew Miller, Jess Harris, and Wendy Taggart. "The impact of COVID-19 on student learning in New South Wales primary schools: an empirical study." *The Australian Educational Researcher* (2021): 1-33.
- * Juniper Education, "The impact of the Covid-19 pandemic on primary school children's learning"
https://21e8jl3324au2z28ej2uho3t-wpengine.netdna-ssl.com/wp-content/uploads/juniper_folder/Juniper-Education-National-Benchmark-Dataset-Report.pdf
- * Kofoed, Michael, Lucas Gebhart, Dallas Gilmore, and Ryan Moschitto. "Zooming to Class?: Experimental Evidence on College Students Online Learning During Covid-19." IZA Discussion Paper
- * Kogan, Vladimir and Stéphane Lavertu, "The COVID-19 Pandemic and Student Achievement on Ohio's Third-Grade English Language Arts Assessment"
http://glenn.osu.edu/educational-governance/reports/reports-attributes/ODE_ThirdGradeELA_KL_1-27-2021.pdf
- * Kuhfeld, Megan Beth Tarasawa, Angela Johnson, Erik Ruzek, and Karyn Lewis, "Learning during COVID-19: Initial findings on students' reading and math achievement and growth"
<https://www.nwea.org/content/uploads/2020/11/Collaborative-brief-Learning-during-COVID-19.NOV2020.pdf>
- * Orlov, George, Douglas McKee, James Berry, Austin Boyle, Thomas DiCiccio, Tyler Ransom, Alex Rees-Jones, and Jörg Stoye. "Learning during the COVID-19 pandemic: It is not who you teach, but how you teach." *Economics Letters* 202 (2021)

- * Pier, Hough, Christian, Bookman, Wilkenfeld, Miller, "Evidence on Learning Loss From the CORE Data Collaborative" <https://edpolicyinca.org/newsroom/covid-19-and-educational-equity-crisis>
- * RS Assessment "The impact of school closures on autumn 2020 attainment" https://www.risingstars-uk.com/media/Rising-Stars/Assessment/RS_Assessment_white_paper_2021_impact_of_school_closures_on_autumn_2020_attainment.pdf
- * Schult, Johannes, and Marlit Annalena Lindner. "Did Students Learn Less During the COVID-19 Pandemic? Reading and Mathematics Competencies Before and After the First Pandemic Wave." <https://psyarxiv.com/pqtgf/>
- * Tomasik, Martin J., Laura A. Helbling, and Urs Moser. "Educational gains of in-person vs. distance learning in primary and secondary schools: A natural experiment during the COVID-19 pandemic school closures in Switzerland." *International Journal of Psychology* (2020).
- * Weidmann, B., Allen, R., Bibby, D., Coe, R., James, L., Plaister, N. and Thomson, D., "Covid-19 disruptions: Attainment gaps and primary school responses," Education Endowment Foundation. https://educationendowmentfoundation.org.uk/public/files/Covid-19_disruptions_attainment_gaps_and_primary_school_responses_-_May_2021.pdf

Reviewer #2:

Remarks to the Author:

This paper analyses the impact of remote learning versus in person learning. To do so, the authors exploit unique variation and data in Brazil. In the context of the COVID-19 pandemic, insights in the difference between both education delivery forms is highly policy relevant. The estimated effects are large, with standardized test scores of 0.32 SD and dropout risks of 365%. I also like that they show that the negative effects are larger if schools did not offer online academic activities prior to the pandemic. In fact, I think that the latter finding deserves even more attention than it received in the submitted version of the paper.

- The authors have unique data, with quarterly standardized test scores in 2020. Unfortunately, there is no information on which schools actually reopened. The authors circumvent this issue by using IV. However, evidence from other countries shows that the leeway that schools receive is used in a non-random way, with school characteristics correlating with the actual reopening. To the extent that in the Brazilian education system more advantaged schools reopened sooner, the estimated effects will be upward biased.
- More attention should be paid to defining the key variables. For example, how is remote learning in Table 1 defined? For the dropout variable in Table 1: what about students without test scores in Q2-Q3, and with a score in Q4, or vice-versa? In table 2, how are 'in person activities' defined? Why is it not possible to define it as a continuous variable for the number of days that in person activities are possible (as municipalities probably did not open simultaneously on the same day)?
- The analysis in Figure 1 is valuable. However, it is counter-intuitive that the risk of dropout is decreasing from grade 9 onwards. I would expect that in the older grades, students would more easily dropout than in the younger years. This might have to do with how the dropout variable is constructed (i.e., a missing test result). Although the supplementary analysis clearly shows that a missing test result is a good predictor for dropout in earlier years, during the pandemic this might be different. More discussion and (anecdotal) evidence would be in place here. Related is the lack of an attrition analysis. This might show whether there is (selective) attrition in the sample (and hence, the dropout).

- For the analysis in Table 2, more discussion and evidence is needed on the characteristics of municipalities that allowed for reopening the schools. This might not be random, but correlated with the socio-economic pattern of the municipality. Although this will be partly captured in the DiD specification, the differences in trend can potentially still result in biased estimates. In Table 2 student characteristics are matched (using what matching method?), but not municipality characteristics. Given that students are non-randomly allocated in municipalities, I would be more interested in the latter.
- It is unclear how the tests are standardized? Did you standardize them by quarter (and if so, how can you compare the estimates without linking questions)? Please discuss this more extensively, as it matters for the internal validity.
- Related to the earlier comment, the authors average math and Portuguese scores because for Q4-2020 only the overall standardized test is available. However, the literature on COVID-19 learning losses shows significant differences between subjects. In the approach taken, the estimates might result in a regression to the mean. Therefore, the authors should also provide estimates (without Q4) for the subjects separately.
- The COVID-19 crisis came as a surprise. In some education systems, there was initially a lack of hardware and software. However, as time passed, education systems could adopt to the new situation. Unfortunately, this might undermine the external validity of the estimates. On the bright side, given that the authors have detailed quarterly data, they could examine how the availability of hardware and software changed the estimated impact of online versus in person learning.
- There are significant differences (even in sign) between the naïve estimates and the DiD estimates in Table 1. Although this is briefly mentioned, a more profound discussion is needed as similar naïve estimates have been used broadly in earlier literature.

Reviewer #3:

Remarks to the Author:

The authors take advantage of a relatively unique situation during Covid, the application every quarter in Sao Paulo Brazil of standardized achievement exams as well as the combination of some in person and some online classes which potentially allows effects of online schooling on learning to be isolated. The authors study both risk of dropout and impacts on learning and find large negative effects on the risk of dropout and on learning during the pandemic. While the topic is of great interest and importance, I have some concerns on the validity of the empirics which I detail below.

1. Defining students to be at risk of dropout if they do not take a quarterly exam applied online during the pandemic strikes me as not very convincing indicator of dropout risk. The authors provide evidence (in the supplementary material) this variable is correlated with actual dropout using pre-pandemic data when students were attending in person classes. I do not believe this is a valid exercise to demonstrate that the same indicator is a predictor of dropout during the pandemic when all school activities are remote. I thus suggest the authors drop this analysis (or call it what it is-probability of not taking the exam) or study the correlation between this variable and returning to school later using actual data from the pandemic on to provide evidence that it effectively measures dropout risk later on e.g. during/after the pandemic.

2. The impacts on learning using the two experiments (e.g. the period of closure to measure reduction in learning and the period when some schools reopen to measure the improvement in learning) have different results by an order of magnitude and this discrepancy casts doubt on what to believe about the true impacts of learning losses. Table 1 (columns 3 to 5) suggests reductions in learning over 9

months of online learning on standardized tests of 0.3 standard deviations whereas Table 2 suggests comparing municipalities where schools returned to those who did not that the return to in person learning led to an increase in test scores of 0.024 standard deviations e.g. less than one tenth the effects implied by Table 1. What are the reasons for this enormous discrepancy and which are we to believe represents the true learning losses due to the closure of schools? The authors need to reconcile these differences and provide guidance to the reader as what the takeaways of the analysis are.

Author Rebuttal to Initial comments

Reviewer 1

Following the worldwide school closures starting in March 2020, there has been an outpouring of studies collecting data on the time use and academic outcomes of children and youth who are kept at home. Many of these have been from Europe and the US, while data from other parts of the world have been lacking. This study presents evidence from Brazil (São Paulo), which offers an important case given the long duration of school closures and the developing country context.

The study is well-designed, using a differences-in-differences design that compares test scores before and after remote learning to corresponding test score growth during the same period in the previous year. It also implements a triple-difference design comparing differences between middle- and high-school students in municipalities where the latter group was allowed to return to school. The study finds large increases in incomplete grades (which they call "dropout risk") and drops in test scores, as well as improvements in test scores among high-school students who were allowed to return.

The study appears to confirm evidence from previous work. The design is close to that of Engzell et al (2021), and better than most other studies. This study is also valuable for estimating treatment effects across the age distribution, across population groups, and by local disease intensity. The finding that disease intensity is mostly unrelated to student outcomes is important since it counters a key objection to studies of learning loss in the wake of COVID: that they reflect wider impacts of the pandemic and not that of school closures as such. The added analysis of the effect of school reopening is well executed and does not have a parallel in existing literature.

My main concern is that the study is not appropriately contextualized with respect to existing studies of COVID-induced learning loss. There are by now dozens of studies on this subject (see bibliography below). Some of them are briefly discussed in the paper, but not until the concluding discussion. The motivation of this paper with respect to other works needs to be made clear already in the introduction. Specifically, the abstract and introduction overclaim when they say that "no study has rigorously documented the educational impacts of remote learning relative to in-person classes within primary and secondary education" and that "the evidence base for the impacts of remote learning is thin".

To my mind, the main contributions of this paper is that it a) uses sound data and methods to expand the evidence on COVID-induced learning loss, b) extends previous evidence centered on Europe and the US to a Latin American country, c) studies population heterogeneity along several dimensions, and d) studies the effect of school re-opening and not just closure.

We really appreciate your summary of the contributions of the paper, and thank you for raising the concern about appropriately contextualizing it relative to previous studies on COVID-induced learning losses. We significantly revised the introduction of the paper following your suggestion. Now, we write (pp.3-4):

"Quantifying learning losses due to remote learning within primary and sec-

ondary education is urgent, as governments need to make informed decisions when trading off the potential health risks of reopening schools in the pandemic⁽¹⁾ against its potential educational benefits. This remains to be the case even with high immunization coverage; in Brazil, while over 45% of the population had received the full course of COVID-19 vaccines by September 2021, only approximately 25% of students had returned to in-person classes until July 2021⁽²⁾. Several papers attempt to quantify learning losses from remote relative to in-person classes before the pandemic, but with important limitations when it comes to generalizability. Most studies are based on developed country settings: some focus on tertiary education, contrasting online to in-person instruction at community colleges⁽³⁾, while those that focus on secondary schools restrict attention to charter schools, contrasting online to in-person instruction within a very selected sets of students^(4–7). In contrast, evidence for developing countries is thinner, and mostly from experiments that use remote learning to expand educational access to rural and remote regions that had no access to education before^(8–11) – a very different counterfactual than in-person classes before the pandemic. In turn, the studies that try to estimate the extent of learning losses due to remote learning during the pandemic either rely on simulations^(12, 13) or suffer from comparability issues, contrasting different tests and student populations before and during the pandemic, without parsing out other direct effects of COVID-19 above and beyond the transition to remote learning^(14–24). Even the few studies that rely on appropriate counterfactuals to study this question have to rest on strong assumptions, given the nature of the variation they use to identify causal effects; in particular, differences in the length of school recess across geographical units or that induced by previous epidemics^(25, 26) are only loosely related to the changes in instruction mode observed in the context of COVID-19. As such, the only credible available evidence for the impacts of remote learning on secondary schools during the pandemic is for developed countries⁽²⁷⁾ – leaving key questions unanswered, from the extent of its impacts on student dropouts (an issue relatively unimportant in developed countries, but critical in the developing world) to the extent of heterogeneity of those impacts with respect to student characteristics, such as age, gender and race.

We contribute to this literature in several respects. First, we take advantage of a natural experiment in São Paulo State to provide estimates of the effects of remote learning under appropriate counterfactuals. In particular, our estimates allow for potential differences between remote and in-person exams, and in the set of students taking them, by leveraging changes in instruction mode between Q1 and Q2–Q4 within the 2020 school year, while assessments were kept fully remote across all 2020 school quarters. An additional natural experiment, linked to staggered school reopening in São Paulo State over the course of Q4/2020, further corroborates the evidence. Second, we provide first-hand evidence on the effects of remote learning in secondary education within developing countries, a context in which most school systems did not even conduct assessments in the context of remote classes⁽²⁸⁾. Third, we are able to parse out the effects of COVID-19 local disease activity to disentangle the effects of remote learning

from other effects of the pandemic on educational outcomes (from health shocks to income losses to effects on students' mental health). Last, we document that the negative impacts of remote learning were larger for girls and for non-white students, and for schools in low-income neighborhoods and those without previous experience with online teaching."

We also excluded from the abstract the sentence that "no study has rigorously documented the educational impacts of remote learning relative to in-person classes within primary and secondary education". We also deleted from the introduction the sentence stating that: "the evidence base for the impacts of remote learning is thin"; instead, now we claim that the conclusions from this literature suffer from generalizability issues (as discussed in detail above). We think that these edits better reflect the current state of the related literature, and thank you for pushing us to characterize it more accurately.

In addition, I have several smaller comments on the analysis and discussion.

More information on the tests needs to be included. How were these designed and for how long did the students sit them? Is there any sense of the reliability? Did the remote testing regime offer any opportunities for cheating? Were the tests taken at Q1 and Q4 identical in design? If not, how was a difference score calculated and what is its interpretation?

We wrote a new section in the Appendix providing additional details on the standardized tests (Supplementary Materials, Appendix C.1, p.5). We write: "The São Paulo State Secretariat (SEDUC) conducts standardized tests (Avaliações de Aprendizagem em Processo, AAPs) with the aim of evaluating students' quarterly progress in Math and Portuguese. Participation in these tests is *not* mandatory, and students who do not participate or those with low scores are not penalized in any way. Having said that, SEDUC strongly incentivizes participation. Schools are required to print materials promoting each test, and to recurrently remind and motivate students to take part in the exam. Such engagement ensured a participation rate of no less than 80% in each and every test throughout 2019 and 2020 – even in those conducted remotely over the course of the pandemic.

The evaluation consists of one math and one Portuguese exam each school quarter. The exams started off as a pilot in 2011, and remained in the same format between 2015 and 2019. Each year, a group of public school teachers is designated to prepare questions for the exams following guidelines on the topics and difficulty level. This is meant to make test scores comparable across years (29). AAPs have been found to contribute to the teaching of Portuguese and to the identification of learning setbacks in specific subjects (30).

In 2020, all exams were applied online (alternatively, students without access to connectivity could fetch printouts at the school gate, and return them the same way). Students had 48 hours to complete the exam. Questions for the exam were prepared the same way as in previous years, except that in 2020 the guidelines for the school curriculum were simplified as soon as it was clear that in-person classes would have to be suspended, to account for the fact that remote learning would not be able to cover as much (31). Exams were applied consistently throughout all schools quarters of 2020, which enables the within-

year comparisons we pursue in the main text.

One important issue is potential cheating in the remote application of the standardized tests. While the Education Secretary had no enforceable mechanism to prevent cheating in remote exams, as discussed above, students had no discernible benefits (costs) from scoring high (low) in the AAPs. Most importantly, as long as cheating is not differential between Q1 and Q2-Q4, it does not affect the comparisons of interests in the main text. Moreover, Appendix C.2 shows that while the distribution of GPA (a key determinant of whether the student graduates or advances to the next grade) considerably changed in 2020 relative to previous years – with significant bunching on minimum passing grades –, the same did *not* happen with the distribution of AAPs' scores, which displays no evidence of bunching neither in previous years nor during the pandemic.”

To better allow comparison with existing work, effect sizes should be discussed in light of the exact length of school closures. What is the expected loss per week of school closure, and what is the expected gain per week of opening up? How does this compare to the estimates reported by Engzell et al (2021) for a European country or Kuhfeld et al for the US?

Thank you for pushing us to present results in a way that helps the reader better contextualize them in the literature. We included these comparisons in the Results section. We now write (p.6): “The average school in São Paulo State remained closed for approximately 35 weeks throughout 2020. As such, our estimates imply that students lost approximately 0.009 standard-deviations of learning each week, relative to in-person classes. This effect size is only slightly higher than that in (27) but at least 4-fold that in (21).” In a footnote (pp.6-7), we explain that “Estimates in (27) imply a learning loss of 0.3 to 0.4 s.d a year, which translates to 0.005-0.007 s.d a week. Estimates in (21) suggest average learning losses of 3-5 percentiles over the course of a year. In order to make our estimates comparable, we ordered test scores in the baseline period (Q4/2019), and evaluated the estimates in Table 1 relative to the score of the median student in the previous year – scaling those losses linearly for one year, so as to keep the time frame constant. Based on those estimates, remote learning would have led to a 22-25 percentile decrease relative to in-person classes – a dramatic effect size.”

Also, when discussing the effects of reopening schools for in-person classes (pp.10-11), we write: “In municipalities that authorized schools to reopen for in-person academic activities in 2020, the average school could have done so for at most 5 weeks. Thus, resuming in-person classes contributed to an increase in test scores of about 0.005 s.d a week. While this effect size is lower than that estimated leveraging school closures, it is actually remarkably high (just as large as weekly learning losses from school closures estimated in (27)) once accounting for the fact that it is based on intention-to-treat (ITT) estimates, as we do not have data on which schools actually reopened (and for how long) in the municipalities that issued authorization decrees.”

Test scores are markedly higher in 2020 compared to 2019. There are several potential reasons for this: sample selection, mode effects on the test

(online/take-home vs in person), and the simplified curriculum during the pandemic. These points appear scattered throughout but need to be brought together at some point in the text. Do the authors view any of them as a more likely explanation?

In the revised manuscript, we discuss this issue more explicitly after Table 1. In page 6, we write: “These naive estimates reflect the fact that average grades are higher in 2020 than in 2019. There are two important factors that likely explain these pattern. First, a selection effect, linked to both higher student dropouts and differential selection into standardized tests during remote learning. This selection effect is what motivate us to implement a matching strategy in column (5). Second, differences in assessment mode between years. In particular, AAPs in 2020 covered a simplified curriculum, and students had much more time to take the exam in 2020 than in previous years (two days, relative to a couple of hours). See Appendix C.1 for a detailed discussion of differences between in-person and remote exams in the context of São Paulo State.”

The main results report differences-in-differences estimates with 2019 as a single comparison year. I assume that the results do not differ much if 2018 is used as a comparison instead, but this information could be included to gauge the robustness. What is the estimate of $(Q4\ 2020-Q1\ 2020)-(Q4\ 2018-Q1\ 2018)$?

Thank you for suggesting this additional robustness check for our results. We now present this estimate in Appendix E (Supplementary Materials, p.12-13). We write: “In Table E.2, we present a slight variation of the main results shown in Table 1, estimating the differences-in-differences model with 2018 instead of 2019 as the counterfactual year. Results are very robust to that alternative definition.”

Table E.2: Effects of remote learning on dropout risk and test scores with alternative baseline period

	(1)	(Q4 2020-Q1 2020)- -(Q4 2018-Q1 2018) (2)	(3)
Panel A: High dropout risk			
Remote learning	0.0655*** (0.0002)	0.0655*** (0.0002)	0.0655*** (0.0002)
Mean 2018 Q4	0.016	0.016	0.016
N		8,312,220	
Panel B: Standardized test scores			
Remote learning	-0.304*** (0.0001)	-0.291*** (0.0001)	-0.310*** (0.0001)
In-person learning equivalent	0.44	0.44	0.44
N		7,001,012	
Grade fixed-effects	yes	yes	yes
Matching	no	yes	yes
Inverse probability weighting	no	no	yes

The introduction and discussion mention that in municipalities in which high school students were allowed back, middle school students also partly returned. This information is not mentioned in the subsection “Effects of Resuming In-person Classe”, so a reader who skips to that section will be puzzled about why attendance increased for both groups in Table 2.

Thank you for pointing that out, and apologies for the omission in the original manuscript. We now make this clearer at the beginning of the section on the effects of resuming in-person classes.

The authors use incomplete grades as a proxy for dropout risk, arguing that enrollment was kept artificially high by school administrators. This seems fine. But, incomplete grades do not appear to be a very good predictor of dropout (Figure A.1). Please report the correlation in this figure. And as a suggestion, why not call the outcome incomplete grades instead?

Motivated by this comment (and similar ones from others referees), we provided more extensive justification for the connection between our proxy and actual dropouts. We start by better motivating this proxy using anecdotal evidence. We write (in p.5): “This proxy has been used for years by the Education Secretary and by philanthropic organizations that support quality education in Brazil to predict student dropouts, especially when it comes to identifying the schools most likely to be affected.” Next, in Section A.1 of the Supplementary Materials, we show that this proxy reliably predicts actual aggregate dropouts in the years before the pandemic. We also collected additional data for the first quarter of 2021, and now show that students who had missing grades in 2020 were much more likely not to have engaged in *any* academic activity in Q1/2021.

Concretely, we write (Supplementary Materials, p.2), when discussing the correlation between our proxy and actual dropouts: “(...) the figure showcases that the classroom-level actual and proxy dropouts are highly correlated, with a coefficient of approximately 0.7. Since measurement error tends to attenuate this correlation, the coefficient represents a lower-bound to the actual prediction power of this proxy. Moreover, actual dropout rates in 2019 were over 6-fold higher among students with missing math and Portuguese grades by the end of the previous school year.”

Last, in Section A.2 of the Supplementary Materials, we write (pp. 2-3):

“The last section shows that, before the pandemic, our proxy reliably predicted actual student dropouts. Nevertheless, its predictive power might have changed during such exceptional times; for instance, many students might have failed to hand in homework and exams in 2020 due to atypical circumstances – e.g., limited connectivity or fear of leaving home to hand them in person – despite no intention of dropping out the following year.

To address this concern, we collected additional administrative data for Q1/2021, when in-person classes had been authorized to return in all municipalities of São Paulo State. As explained in the main text, all students who had not yet graduated from high school were automatically enrolled in 2021; hence, there is no data on actual dropouts for 2020. Instead, we focus on whether students engaged in *any* academic activity during this school quarter: attending classes, handing in assignments or taking exams that would qualify for scorecard grades in Q1/2021, across all school subjects. If our proxy still predicts actual dropouts in 2020, we would expect that students at high dropout risk during that year are less likely to participate in any academic activity in 2021. As such, we compute the odd ratio of participating in academic activities for students classified as high risk of dropout or not.

We find that students with missing grades in the last quarter of 2020 were 8.6 times more likely not to have attended a single class, and 9.7 times more likely not to have taken a single test in Q1/2021. Hence, we conclude that our proxy remains a reliable predictor of dropout risk even throughout the pandemic.”

Nevertheless, we added nuance to our discussion of the potential caveats of relying on this proxy. We write (in p.10): “Will students that did not engage with schools activities in the absence of in-person classes remain permanently out of the school system? Worryingly, we find that students at high dropout risk in 2020 are almost 10 times more likely to continue not engaged with schools activities even as in-person classes returned in 2021, suggesting that this is not a transitory phenomenon. Having said that, it might be too early to answer that question. Whether the high risk of student dropouts will ultimately materialize depends on public policy responses – from engaging students’ families (32) to creating additional (possibly financial) incentives for secondary students to remain in school.”

Figure B.1 suggests that most of the shortfall in test score growth occurred between Q1 and Q2 when classes were supposedly still in person, and rates of learning were similar thereafter. Does this affect conclusions about the efficacy of remote learning? Should we conclude that educators and families struggled

initially but eventually adapted with time?

We thank you for this excellent observation, which we had not appropriately discussed in the previous version of the manuscript. We now address the dynamic pattern of treatment effects in the Discussion section of the paper. We write (in p.9): “The lion’s share of the impacts on test scores takes place over the first quarter of remote classes. This might lead to the concern that learning losses are not the result of remote learning, but rather of the transition from in-person to remote. If that were the case, then we should expect that, by the end of the school year, when students and teachers had the time to adapt to the new instruction mode, the effectiveness of remote learning would converge to that of in-person classes. This hypothesis is, however, inconsistent with two key findings of the paper. First, dropout risk surges after the Q3, inconsistent with the idea of a transitory shock. Second, the return of in-person activities in November, already 8 months into remote learning, significantly increases students’ test scores; as discussed, the effect size is very large, especially before the facts that schools were open for a short period of time, and that we can only estimate intention-to-treat effects. As such, our interpretation is that the early onset of learning losses is rather linked to non-linear treatment effects: the magnitude of learning losses is so large at Q2 that it is simply not possible that the test scores keep deteriorating at that same pace indefinitely.”

Figure C.1-C.2. It seems unconventional to use a continuous density function to represent a discrete outcome. These graphs could be made clearer if a histogram was used.

We thank you for this suggestion. Actually, the GPA distribution in Figure C1 is approximately continuous in the interval $[0,10]$; we apologize for not making it clearer in the original manuscript. Using a continuous density to approximate the GPA distribution also allows us to better highlight the bunching at minimum passing grades in 2020, in Figure C1. In turn, while standardized test scores depicted in Figure C2 are computed as the normalized % of correct answers – indeed, a discrete variable –, the range of possible values is very large. We experimented with alternative visualizations, including histograms with bins for each possible value, but the version included in the manuscript remained our favorite. Having said that, if you would like us to include the discrete visualization in the revised manuscript we would be happy to do so.

In the analysis of heterogeneous treatment effects by student demographics shown in Appendix D, no coefficients or SEs are reported. There is little way to gauge the magnitude of these differences and their significance. Adding these numbers would be helpful.

We thank you for raising this issue, which has led us to completely re-write Appendix D for ease of visualization and interpretation. Instead of only depicting effect sizes relative to initial conditions, the updated figures now showcase the evolution of the outcomes over time for each sub-group. We also included a new table with estimated coefficients and standard-errors. Now, we write (Supplementary Materials, pp. 8-10): “Figures S.D.1 and S.D.2 showcase average standardized test scores and dropout risk, respectively, by school quarter of 2019 and 2020 and by selected student and school characteristics.

Figure S.D.1: Average standardized test scores, by school quarter and sub-group

Figure S.D.2: Average high dropout risk, by school quarter and sub-group

Table S.D.1 reports differences-in-differences estimates of the impacts of remote learning for each sub-group.

Table S.D.1: Heterogeneous treatment effects of remote learning on test scores and dropout risk

Group	Treatment effects	
	Std. test scores	High dropout risk
Male	-0.321*** (0.0006)	0.060*** (0.0003)
Female	-0.321*** (0.0006)	0.071*** (0.0003)
p-value difference	0.54	0.00
White	-0.289*** (0.0006)	0.068*** (0.0003)
Non-white	-0.344*** (0.0009)	0.081*** (0.0005)
p-value difference	0.00	0.00
High income	-0.291*** (0.0006)	0.054*** (0.0003)
Low income	-0.420*** (0.0006)	0.091*** (0.0003)
p-value difference	0.00	0.00
Online	-0.271*** (0.0006)	0.053*** (0.0004)
Not online	-0.431*** (0.0006)	0.090*** (0.0003)
p-value difference	0.00	0.00

While Figure S.D.1 suggests that girls – whose average test scores were higher than those of boys by Q4/2019, but lower by Q4/2020 – were hurt disproportionately by remote learning, Table S.D.1 shows that this is actually *not* the case, at least when it comes to test scores. That false impression stems from naive comparisons of test scores coming from different assessment modes; in effect, the differences-in-differences estimate documents that remote learning hurt boys’ and girls’ test scores by exactly the same extent. When it comes to dropout risk, however, girls were really hurt to a greater extent (a nearly 18% higher effect size, significant at the 1% level). Table S.D.1 also shows that, relative to white students, non-whites experienced 19% larger losses in test scores due to remote learning ($p=0.00$) and a 19% larger increase in dropout risk ($p=0.00$). Strikingly, learning losses due to remote learning were much more dramatic in low-income schools: relative to schools located in above-median per capita income neighborhoods, the former experienced a 44% larger impacts on standardized test scores ($p=0.00$) and a 69% larger increase in dropout risk ($p=0.00$). Similarly, schools without online academic activities prior to the pan-

demographic experienced 59% larger impacts on standardized test scores ($p=0.00$) and a 70% larger increase in dropout risk ($p=0.00$).

These results are important enough that we added them to the introduction of the revised manuscript. We now write (in p.2) “Learning losses were significantly larger among non-white students, and within vulnerable schools located in low-income neighborhoods and without prior experience with online academic activities”.

Table E.1 reveals that non-white, poorer and low-performing students are underrepresented in 2020. The authors use propensity score reweighting to address this but confounding on unobservables might remain. However, the fact that adjustment for observed confounders does not alter estimates much between column (3) and (4)-(5) in Table 1 can be marshalled to claim that residual confounding is also likely to be small.

We thank you for raising that issue. We have now included an observation that explicitly makes this point. Specifically, we write in p.6: “In Panel B, columns (4) and (5), we use propensity score matching to account for potential selection of students into standardized tests based on characteristics, especially given the significant treatment effects of remote learning on dropout risk documented in Panel A. Results are very robust to the matching procedure. This is not because selection is unimportant – Table S.E.1 shows that student characteristics matter significantly for differential take-up of standardized tests in 2020 –; rather, this presumably largely reflects the fact that the *nature of selection does not change* between Q1 and Q2-Q4/2020. As such, self-selection into the exams has small to no impacts on our main results.”

A key contribution of this paper is to study heterogeneity by local COVID incidence. The authors write that “risk increased with local disease activity” but Figure F.1 largely looks like the absence of a meaningful association to me.

We agree with that conclusion. The way we framed this in the original manuscript was conservative, based on the fact that our non-parametric estimate of the effects of remote learning on dropout risk increases with local disease activity in Panel A of Figure S.F.1; however, as you correctly point out, this increase is not statistically significant. We have changed the text to reflect this comment; specifically, we now write (in p.5): “We refine our differences-in-differences strategy for the effects of remote learning on test scores by matching student characteristics across years, to parse out the effects of selection. We estimate a propensity score within each grade and quarter, based on student characteristics (see section E of the Supplementary Materials), and control flexibly for it (with a cubic polynomial) to parse out selection. We also use these propensity scores to re-weight observations (by the inverse of their selection probability) to ensure that treatment effects on standardized test scores reflect those on the universe of students in each grade. We allow our estimates to vary non-parametrically with municipal-level per capita COVID-19 cases over that period, to gauge the magnitude of treatment effects in the absence of other effects of disease activity on learning outcomes and find losses did not vary systematically with local disease activity.”

However, the detected case load depends on testing capacity. If learning loss

is larger in poorer communities and testing capacity correlates negatively with poverty, null effects might be spurious. Is this a worry?

Thank you for raising this potentially important concern. To test whether under-reporting could drive our null results, we replicated the non-parametric analyses using COVID-19 deaths rather than cases as our measure of local disease activity. Deaths are much less likely to be under-notified than cases (see the discussion in (1)), and are much less likely to merely reflect local testing capacity. Figure F.2, included below for your convenience, shows very similar patterns for our results: non-parametric treatment effects do not increase with local COVID-19 deaths either. These results are included in Appendix F (Supplementary Materials, p.18):

Figure S.F.2: Non-parametric heterogeneous treatment effects on educational outcomes by local COVID-19 deaths

In referring to Supplementary Materials, please state the specific Figure/Table you are referring to.

We apologize for the imprecision in the original manuscript. We now make explicit reference to the specific sections of the Supplementary Materials we allude to in the main text. We thank you for all your comments and suggestions, which we think have helped us greatly improve the paper.

Reviewer 2

This paper analyses the impact of remote learning versus in person learning. To do so, the authors exploit unique variation and data in Brazil. In the context of the COVID-19 pandemic, insights in the difference between both education delivery forms is highly policy relevant. The estimated effects are large, with standardized test scores of 0.32 SD and dropout risks of 365%. I also like that they show that the negative effects are larger if schools did not offer online academic activities prior to the pandemic. In fact, I think that the latter finding deserves even more attention than it received in the submitted version of the paper.

Thank you for this suggestion and for the attention you have given to the paper. We now give more emphasis to heterogeneous treatment effects in the introduction of the paper. Specifically, we now write (in p.4) “Last, we focus a great deal on heterogeneous treatment effects, illustrating how the impacts of remote learning on test scores and dropout risk vary by students’ age, gender and race, and by school’s socioeconomic status and access to online activities prior to the pandemic.”

The authors have unique data, with quarterly standardized test scores in 2020. Unfortunately, there is no information on which schools actually reopened. The authors circumvent this issue by using IV. However, evidence from other countries shows that the leeway that schools receive is used in a non-random way, with school characteristics correlating with the actual reopening. To the extent that in the Brazilian education system more advantaged schools reopened sooner, the estimated effects will be upward biased.

Thank you for raising this important concern. To clarify, we cannot estimate IV because we do not observe which schools actually reopened (and for how long). As such, we can only estimate intention-to-treat (ITT) effects, based on municipal-level authorization decrees. The decision to reopen schools was undertaken at the municipal level. Since primary and secondary education is mandatory in Brazil, there is ultimately no leeway for school-level decisions beyond what is determined by municipal decrees. In any case, municipalities that authorized schools to reopen in 2020 had indeed different characteristics; in particular, they had higher per capita income than those that did not (see discussion below). Nevertheless, any such differences should not bias the results in Table 2; the reason is that the differences-in-differences strategy parses out the effects of any municipality, school or student characteristics that do not change over time. Importantly, in a companion paper (*1*), we document that the decision to reopen schools was also uncorrelated with the recent local evolution of COVID-19 cases or deaths. While the differences-in-differences strategy cannot parse out the effects of other characteristics that changed differentially over time across municipalities that decided to reopen schools and those that did not, we additionally undertake a triple-differences analysis, comparing different students *within each municipality*, taking advantage of the fact that in-person classes returned in 2020 only for high-school students, but not for middle-school ones. Panel C of Table 2 shows that treatment effects on test scores are entirely

driven by high-school students within municipalities that authorized schools to reopen, confirming that our estimates capture the effects of in-person classes rather than other differences in municipal characteristics.

More attention should be paid to defining the key variables. For example, how is remote learning in Table 1 defined? For the dropout variable in Table 1: what about students without test scores in Q2-Q3, and with a score in Q4, or vice-versa? In table 2, how are 'in person activities' defined? Why is it not possible to define it as a continuous variable for the number of days that in person activities are possible (as municipalities probably did not open simultaneously on the same day)?

We apologize for the imprecision in the previous version of the manuscript. We have now included detailed descriptions of all key variables in the revised manuscript. Specifically, we write (in p.6): " We define high dropout risk as equal to 1 if a student had no math and no Portuguese grades on record in that school quarter, and 0 otherwise. (...) When it comes to standardized test scores, we average math and Portuguese scores within each quarter (for Q4-2020, only overall standardized test scores are available). We keep in the sample only students with valid test scores in Q1 and Q4 of each school year. Our main treatment variable indicates whether students were exposed to remote rather than in-person classes; as such, it equals 1 for Q2-Q4/2020, and 0 otherwise."

Also, we write (in p.8): "In this section, we examine a second natural experiment. From October 2020 onwards, some municipalities allowed schools to reopen for in-person activities, following safe reopening protocols. In schools that actually reopened, from November 2020 onwards high-school classes were allowed to resume in-person; in contrast, middle-school classes remained completely remote until the end of 2020. That motivates the additional differences-in-differences estimates, and the triple-differences strategy that we pursue in this section. (...) where in-person activities is an indicator variable equal to 1 if a municipality authorized schools to reopen, and 0 otherwise".

When it comes to the definition of in-person activities, your comment is correct: we now estimate an alternative version of Table 2, with a continuous variable indicating the number of weeks that schools were authorized to reopen for in-person activities in 2020 instead. While this is not our favorite specification, since the length of actual reopening is measured with error (see (1) for further details), Table S.E.4 shows that the pattern of results are very robust to this alternative definition of treatment for in-person classes. We now write (in pp. 8): "In municipalities that authorized schools to reopen for in-person academic activities in 2020, the average school could have done so for at most 5 weeks. Thus, resuming in-person classes contributed to an increase in test scores of about 0.005 s.d a week. While this effect size is lower than that estimated leveraging school closures, it is actually remarkably high (just the same as in (27)) once accounting for the fact that it is based on intention-to-treat (ITT) estimates, as we do not have data on which schools actually reopened (and for how long) in the municipalities that issued authorization decrees."

Table S.E.4: ITT effects of resuming in-person school activities on dropout risk and test scores with continuous treatment

	(1) Attendance	(2) Std. test scores	(3) Dropout risk
Panel A: Diff-in-diff: Middle school			
In-person activities	0.001 (0.001)	0.002 (0.001)	0.0002 (0.001)
Panel B: Diff-in-diff: High school			
In-person activities	0.0001 (0.001)	0.005*** (0.0002)	0.0003 (0.0002)
Panel C: Triple differences			
In-person activities	-0.00001 (0.002)	0.003*** (0.0001)	0.0001 (0.0001)
Grade fixed-effects	yes	yes	yes
Matching	yes	yes	yes
N	3,701,482	2,624,943	3,701,482

The analysis in Figure 1 is valuable. However, it is counter-intuitive that the risk of dropout is decreasing from grade 9 onwards. I would expect that in the older grades, students would more easily dropout than in the younger years. This might have to do with how the dropout variable is constructed (i.e., a missing test result). Although the supplementary analysis clearly shows that a missing test result is a good predictor for dropout in earlier years, during the pandemic this might be different. More discussion and (anecdotal) evidence would be in place here. Related is the lack of an attrition analysis. This might show whether there is (selective) attrition in the sample (and hence, the dropout).

Motivated by this comment (and similar ones from others referees), we provided more extensive justification for the connection between our proxy and actual dropouts. We start by better motivating this proxy using anecdotal evidence. We write (in p.5): “This proxy has been used for years by the Education Secretary and by philanthropic organizations that support quality education in Brazil to predict student dropouts, especially when it comes to identifying the schools most likely to be affected.” Next, in Section A.1 of the Supplementary Materials, we show that this proxy reliably predicts actual aggregate dropouts in the years before the pandemic. We also collected additional data for the first quarter of 2021, and now show that students who had missing grades in 2020 were much more likely not to have engaged in *any* academic activity in Q1/2021.

Concretely, we write (Supplementary Materials, p.2), when discussing the correlation between our proxy and actual dropouts: “(...) the figure showcases that the classroom-level actual and proxy dropouts are highly correlated, with a coefficient of approximately 0.7. Since measurement error tends to attenuate this correlation, the coefficient represents a lower-bound to the actual prediction

power of this proxy. Moreover, actual dropout rates in 2019 were over 6-fold higher among students with missing math and Portuguese grades by the end of the previous school year.”

Last, in Section A.2 of the Supplementary Materials, we write (pp. 2-3):

“The last section shows that, before the pandemic, our proxy reliably predicted actual student dropouts. Nevertheless, its predictive power might have changed during such exceptional times; for instance, many students might have failed to hand in homework and exams in 2020 due to atypical circumstances – e.g., limited connectivity or fear of leaving home to hand them in person – despite no intention of dropping out the following year.

To address this concern, we collected additional administrative data for Q1/2021, when in-person classes had been authorized to return in all municipalities of São Paulo State. As explained in the main text, all students who had not yet graduated from high school were automatically enrolled in 2021; hence, there is no data on actual dropouts for 2020. Instead, we focus on whether students engaged in *any* academic activity during this school quarter: attending classes, handing in assignments or taking exams that would qualify for scorecard grades in Q1/2021, across all school subjects. If our proxy still predicts actual dropouts in 2020, we would expect that students at high dropout risk during that year are less likely to participate in any academic activity in 2021. As such, we compute the odd ratio of participating in academic activities for students classified as high risk of dropout or not.

We find that students with missing grades in the last quarter of 2020 were 8.6 times more likely not to have attended a single class, and 9.7 times more likely not to have taken a single test in Q1/2021. Hence, we conclude that our proxy remains a reliable predictor of dropout risk even throughout the pandemic.”

When it comes to the pattern of heterogeneous treatment effects depicted in Figure 1, it turns out that estimated treatment effects are actually higher for high-school students (in percentage points), in line with your intuition. Figure 1, however, expresses effect sizes as a proportion of average baseline dropout risk, which was much higher for high-school students (2-3-fold that for middle-school students) in 2019. We added this argument in p.7 of the revised manuscript).

When it comes to attrition, for the analysis of treatment effects on dropout risk we have data on the universe of students from São Paulo State secondary schools. This is why the analyses in Panel A of Table 1 and Panel A of Figure 1 require no statistical correction for sample selection. In turn, attrition potentially matters for our estimates of treatment effects on standardized test scores, since we only observe that outcome for the students who ultimately take the exams. This is why we rely on propensity score matching in columns (4) and (5) of Table 1. We explicitly discuss these issues in the revised version of the manuscript. Specifically, we write in p.6-7: “In Panel B, columns (4) and (5), we use propensity score matching to account for potential selection of students into standardized tests based on characteristics, especially given the significant treatment effects of remote learning on dropout risk documented in Panel A. Results are very robust to the matching procedure. This is not because selection is unimportant – Table S.E.1 shows that student characteristics matter

significantly for differential take-up of standardized tests in 2020 –; rather, this presumably largely reflects the fact that the nature of selection does not change between Q1 and Q2-Q4/2020. As such, self-selection into the exams has small to no impacts on our main results.”

For the analysis in Table 2, more discussion and evidence is needed on the characteristics of municipalities that allowed for reopening the schools. This might not be random, but correlated with the socio-economic pattern of the municipality. Although this will be partly captured in the DiD specification, the differences in trend can potentially still result in biased estimates. In Table 2 student characteristics are matched (using what matching method?), but not municipality characteristics. Given that students are non-randomly allocated in municipalities, I would be more interested in the latter.

Thank you again for raising this important concern. We included an additional Appendix to provide details on the school reopening process. In Appendix G of the supplementary materials, we now write: “As discussed in the main text, some schools partially reopened to in-person activities at the end of 2020. In this section, we provide additional details on the reopening process. São Paulo State authorized municipalities to reopen schools for optional activities (remedial classes for students lagging behind, and extra-curricular activities, such as psychological counselling) from September 8th to high-school students, and from October 7th to primary- and middle-school students. Regular in-person classes for high-school students were authorized to return from November 3rd. Only municipalities within health regions with stable pandemic conditions were allowed to return.

Municipalities had autonomy to decree whether schools could reopen, as long as safe reopening protocols were in place; in particular, all school staff had to wear personal protective equipment, alcohol had to be made available at the school gate, and attendance was limited (e.g. at 35% capacity in regions where the severity of the pandemic was high). The State Secretariat of Education estimates that 1,700 schools were in fact open for in-person activities and that at least 2 million students did go to school during that period.

The reopening process was staggered across municipalities. Figure S.G.1 shows the cumulative distribution of municipalities which authorized schools to reopen over time.

Table G.1: Cumulative distribution of school reopening authorization decrees in São Paulo State
Naturally, the decision to reopen schools for in-person activities was not randomly assigned. External conditions mattered; in particular, municipalities in health regions with high disease activity *could not* issue authorization decrees before reaching a low enough threshold for COVID-19 cases and in-patient hospitalizations. Moreover, even among those that could, municipalities that authorized schools to reopen were not identical to those that did not. Table G.1 provides descriptive statistics of the municipalities that reopened schools and those that did not. Municipalities that reopened schools had a lower number of new COVID-19 cases and deaths, were relatively poorer, and less populous”.

Table S.G.1: Descriptive statistics in the baseline (end of September)

	Never treated	Ever treated	p-value difference
New cases per thousand	0.79	0.76	0.71
New deaths per thousand	0.03	0.02	0.23
Accumulated deaths per thousand	0.44	0.49	0.20
Income per capita	672.17	804.58	0.00
Population (thousands)	38.65	200.37	0.09
Number of schools	19.41	67.94	0.03
Number of students (thousands)	7.21	34.34	0.55
School infrastructure	-0.01	0.00	0.88
Municipalities	514	129	

Nevertheless, any such differences should not bias the results in Table 2; the reason is that the differences-in-differences strategy parses out the effects of any municipality, school or student characteristics that do not change over time. Importantly, in a companion paper (1), we document that the decision to reopen schools was also uncorrelated with the recent local evolution of COVID-19 cases or deaths. While the differences-in-differences strategy cannot parse out

the effects of other characteristics that changed differentially over time across municipalities that decided to reopen schools and those that did not, we additionally undertake a triple-differences analysis, comparing different students *within each municipality*, taking advantage of the fact that in-person classes returned in 2020 only for high-school students, but not for middle-school ones. Panel C of Table 2 shows that treatment effects on test scores are entirely driven by high-school students within municipalities that authorized schools to reopen, confirming that our estimates capture the effects of in-person classes rather than other differences in municipal characteristics.

As an additional robustness check, we also replicated the differences-in-differences estimates in Table 2 while matching observations based on municipality-level characteristics. Results in Table S.E.5 are robust to that alternative matching procedure.

Table S.E.5: ITT effects of in-person school activities on dropout risk and test scores matching on municipal characteristics

	(1) Attendance	(2) Std. test scores	(3) Dropout risk
Panel A: Diff-in-diff: Middle school			
In-person activities	0.009*** (0.001)	0.001 (0.001)	0.001 (0.001)
Panel B: Diff-in-diff: High school			
In-person classes	0.008*** (0.001)	0.021*** (0.0001)	0.002 (0.002)
Grade fixed-effects	yes	yes	yes
Municipality matching	yes	yes	yes
N	3,701,482	2,624,943	3,701,482

- It is unclear how the tests are standardized? Did you standardize them by quarter (and if so, how can you compare the estimates without linking questions)? Please discuss this more extensively, as it matters for the internal validity.

We apologize for not providing a more detailed explanation in the original manuscript. In the revised manuscript we provide a much more extensive discussion on the standardized tests. In Appendix C.1 (Supplementary Materials, p.5) we write: “The São Paulo State Secretariat (SEDUC) conducts standardized tests (Avaliações de Aprendizagem em Processo, AAPs) with the aim of evaluating students’ quarterly progress in Math and Portuguese. Participation in these tests is *not* mandatory, and students who do not participate or those with low scores are not penalized in any way. Having said that, SEDUC strongly incentivizes participation. Schools are required to print materials promoting each test, and to recurrently remind and motivate students to take part in the exam. Such engagement ensured a participation rate of no less than 80% in each and every test throughout 2019 and 2020 – even in those conducted remotely over

the course of the pandemic.

The evaluation consists of one math and one Portuguese exam each school quarter. The exams started off as a pilot in 2011, and remained in the same format between 2015 and 2019. Each year, a group of public school teachers is designated to prepare questions for the exams following guidelines on the topics and difficulty level. This is meant to make test scores comparable across years (29). AAPs have been found to contribute to the teaching of Portuguese and to the identification of learning setbacks in specific subjects (30).

In 2020, all exams were applied online (alternatively, students without access to connectivity could fetch printouts at the school gate, and return them the same way). Students had 48 hours to complete the exam. Questions for the exam were prepared the same way as in previous years, except that in 2020 the guidelines for the school curriculum were simplified as soon as it was clear that in-person classes would have to be suspended, to account for the fact that remote learning would not be able to cover as much (31). Exams were applied consistently throughout all schools quarters of 2020, which enables the within-year comparisons we pursue in the main text.

One important issue is potential cheating in the remote application of the standardized tests. While the Education Secretary had no enforceable mechanism to prevent cheating in remote exams, as discussed above, students had no discernible benefits (costs) from scoring high (low) in the AAPs. Most importantly, as long as cheating is not differential between Q1 and Q2-Q4, it does not affect the comparisons of interests in the main text. Moreover, Appendix C.2 shows that while the distribution of GPA (a key determinant of whether the student graduates or advances to the next grade) considerably changed in 2020 relative to previous years – with significant bunching on minimum passing grades –, the same did *not* happen with the distribution of AAPs' scores, which displays no evidence of bunching neither in previous years nor during the pandemic⁷.

In Appendix C.1, we emphasize that the AAPs were built to allow exactly this type of comparisons. We standardize grades relative to the baseline level only in order to express effect sizes in standard deviations, as typically done in the Education literature.

Related to the earlier comment, the authors average math and Portuguese scores because for Q4-2020 only the overall standardized test is available. However, the literature on COVID-19 learning losses shows significant differences between subjects. In the approach taken, the estimates might result in a regression to the mean. Therefore, the authors should also provide estimates (without Q4) for the subjects separately.

Thank you so much for pushing us to estimate impacts separately for math and Portuguese test scores. In the revised manuscript, we now estimate treatment effects of remote learning separately for math and Portuguese grades based off differences between Q1 and Q2-Q3 within 2020 relative to those within 2019. Table S.E.3 shows that effect sizes are larger for math than Portuguese. Importantly, results are inconsistent with regression to the mean.

Table S.E.3: Effects of remote learning on test scores, separately for Portuguese and math

	(Q3 2020-Q1 2020)	(Q3 2020-Q1 2020)-(Q3 2019-Q1 2019)	(Q3 2019-Q1 2019)
	(1)	(2)	(3)
Panel A: Portuguese test scores			
Remote learning	-0.255*** (0.0002)	-0.265*** (0.0002)	-0.267*** (0.0002)
In-person learning equivalent	0.44	0.44	0.44
N		7,131,922	
Panel B: Math test scores			
Remote learning	-0.361*** (0.0002)	-0.342*** (0.0002)	-0.355*** (0.0002)
In-person learning equivalent	0.44	0.44	0.44
N		7,131,922	
Grade fixed-effects	yes	yes	yes
Matching	no	yes	yes
Inverse probability weighting	no	no	yes

“Table S.E.3 documents that learning losses due to remote learning are massive for both subjects, but especially so for math. While students learned only 40% of they would have learned under in-person classes in Portuguese, that figure was only 20% in math classes.”

The COVID-19 crisis came as a surprise. In some education systems, there was initially a lack of hardware and software. However, as time passed, education systems could adopt to the new situation. Unfortunately, this might undermine the external validity of the estimates. On the bright side, given that the authors have detailed quarterly data, they could examine how the availability of hardware and software changed the estimated impact of online versus in-person learning.

Thank you for raising this important concern. The data we have on online access points out that over the course of 2020, only roughly 10% of secondary school students were following remote classes online in São Paulo States; most students actually experienced remote instruction through television (broadcasted centrally and, hence, uniformly across schools). We unfortunately do not have data on school-specific hardware utilization or availability. The best we can do is explore heterogeneity in the extent to which prior engagement in online academic activities moderated the educational impacts of remote learning, according to the 2019 Educational Census. As the paper documents, effects were in fact concentrated in schools without prior online engagement.

Nevertheless, in the revised manuscript we are now better able to shed light on the short and long-run effects of remote learning. We consider this discussion to be important because it distinguishes the learning losses generated by remote classes more broadly from transitional costs linked to the surprise of the pandemic and adaptation costs. We now address the dynamic pattern of treatment

effects in the Discussion section of the paper. We write (in p.8): “The lion’s share of the impacts on test scores takes place over the first quarter of remote classes. This might lead to the concern that learning losses are not the result of remote learning, but rather of the transition from in-person to remote. If that were the case, then we should expect that, by the end of the school year, when students and teachers had the time to adapt to the new instruction mode, the effectiveness of remote learning would converge to that of in-person classes. This hypothesis is, however, inconsistent with two key findings of the paper. First, dropout risk surges after the Q3, inconsistent with the idea of a transitory shock. Second, the return of in-person activities in November, already 8 months into remote learning, significantly increases students’ test scores; as discussed, the effect size is very large, especially before the facts that schools were open for a short period of time, and that we can only estimate intention-to-treat effects. As such, our interpretation is that the early onset of learning losses is rather linked to non-linear treatment effects: the magnitude of learning losses is so large at Q2 that it is simply not possible that the test scores keep deteriorating at that same pace indefinitely.”

There are significant differences (even in sign) between the naïve estimates and the DiD estimates in Table 1. Although this is briefly mentioned, a more profound discussion is needed as similar naïve estimates have been used broadly in earlier literature.

Thank you for pointing that out. In the revised manuscript, we discuss this issue more explicitly after Table 1. In page 6, we write: “These naïve estimates reflect the fact that average grades are higher in 2020 than in 2019. There are two important factors that likely explain these pattern. First, a selection effect, linked to both higher student dropouts and differential selection into standardized tests during remote learning. This selection effect is what motivate us to implement a matching strategy in column (5). Second, differences in assessment mode between years. In particular, AAPs in 2020 covered a simplified curriculum, and students had much more time to take the exam in 2020 than in previous years (two days, relative to a couple of hours). See Appendix C.1 for a detailed discussion of differences between in-person and remote exams in the context of São Paulo State.”

Reviewer 3

The authors take advantage of a relatively unique situation during COVID, the application every quarter in Sao Paulo Brazil of standardized achievement exams as well as the combination of some in person and some online classes which potentially allows effects of online schooling on learning to be isolated. The authors study both risk of dropout and impacts on learning and find large negative effects on the risk of dropout and on learning during the pandemic. While the topic is of great interest and importance, I have some concerns on the validity of the empirics which I detail below.

1. Defining students to be at risk of dropout if they do not take a quarterly exam applied online during the pandemic strikes me as not very convincing indicator of dropout risk. The authors provide evidence (in the supplementary material) this variable is correlated with actual dropout using pre-pandemic data when students were attending in person classes. I do not believe this is a valid exercise to demonstrate that the same indicator is a predictor of dropout during the pandemic when all school activities are remote. I thus suggest the authors drop this analysis (or call it what it is-probability of not taking the exam) or study the correlation between this variable and returning to school later using actual data from the pandemic on to provide evidence that it effectively measures dropout risk later on e.g. during/after the pandemic.

Motivated by this comment (and similar ones from others referees), we provided more extensive justification for the connection between our proxy and actual dropouts. We start by better motivating this proxy using anecdotal evidence. We write (in p.5): “This proxy has been used for years by the Education Secretary and by philanthropic organizations that support quality education in Brazil to predict student dropouts, especially when it comes to identifying the schools most likely to be affected.” Next, in Section A.1 of the Supplementary Materials, we show that this proxy reliably predicts actual aggregate dropouts in the years before the pandemic. We also collected additional data for the first quarter of 2021, and now show that students who had missing grades in 2020 were much more likely not to have engaged in *any* academic activity in Q1/2021.

Concretely, we write (Supplementary Materials, p.2), when discussing the correlation between our proxy and actual dropouts: “(...) the figure showcases that the classroom-level actual and proxy dropouts are highly correlated, with a coefficient of approximately 0.7. Since measurement error tends to attenuate this correlation, the coefficient represents a lower-bound to the actual prediction power of this proxy. Moreover, actual dropout rates in 2019 were over 6-fold higher among students with missing math and Portuguese grades by the end of the previous school year.”

Last, in Section A.2 of the Supplementary Materials, we write (pp. 2-3):

“The last section shows that, before the pandemic, our proxy reliably predicted actual student dropouts. Nevertheless, its predictive power might have changed during such exceptional times; for instance, many students might have failed to hand in homework and exams in 2020 due to atypical circumstances

– e.g., limited connectivity or fear of leaving home to hand them in person – despite no intention of dropping out the following year.

To address this concern, we collected additional administrative data for Q1/2021, when in-person classes had been authorized to return in all municipalities of São Paulo State. As explained in the main text, all students who had not yet graduated from high school were automatically enrolled in 2021; hence, there is no data on actual dropouts for 2020. Instead, we focus on whether students engaged in *any* academic activity during this school quarter: attending classes, handing in assignments or taking exams that would qualify for scorecard grades in Q1/2021, across all school subjects. If our proxy still predicts actual dropouts in 2020, we would expect that students at high dropout risk during that year are less likely to participate in any academic activity in 2021. As such, we compute the odd ratio of participating in academic activities for students classified as high risk of dropout or not.

We find that students with missing grades in the last quarter of 2020 were 8.6 times more likely not to have attended a single class, and 9.7 times more likely not to have taken a single test in Q1/2021. Hence, we conclude that our proxy remains a reliable predictor of dropout risk even throughout the pandemic.”

Nevertheless, we added nuance to our discussion of the potential caveats of relying on this proxy. We write (in p.10): “Will students that did not engage with schools activities in the absence of in-person classes remain permanently out of the school system? Worryingly, we find that students at high dropout risk in 2020 are almost 10 times more likely to continue not engaged with schools activities even as in-person classes returned in 2021, suggesting that this is not a transitory phenomenon. Having said that, it might be too early to answer that question. Whether the high risk of student dropouts will ultimately materialize depends on public policy responses – from engaging students’ families (32) to creating additional (possibly financial) incentives for secondary students to remain in school.”

2. The impacts on learning using the two experiments (e.g. the period of closure to measure reduction in learning and the period when some schools reopen to measure the improvement in learning) have different results by an order of magnitude and this discrepancy casts doubt on what to believe about the true impacts of learning losses. Table 1 (columns 3 to 5) suggests reductions in learning over 9 months of online learning on standardized tests of 0.3 standard deviations whereas Table 2 suggests comparing municipalities where schools returned to those who did not that the return to in person learning led to an increase in test scores of 0.024 standard deviations e.g. less than one tenth the effects implied by Table 1. What are the reasons for this enormous discrepancy and which are we to believe represents the true learning losses due to the closure of schools? The authors need to reconcile these differences and provide guidance to the reader as what the takeaways of the analysis are.

This is, in fact, an important observation that we did not appropriately discuss in the previous version of the manuscript. We apologize for that omission. This difference in effect sizes directly reflects difference in the duration and intensity of each shock. When it comes to duration, the period during which

schools remained closed in 2020 was much longer than that over which some of them reopened for in-person activities. All in-person academic activities were suspended at the end of March 2020, and remained so until the first week of October. As such, the average school in São Paulo was closed for approximately 35 weeks during the year. In contrast, conditional on reopening, the average school resumed in-person classes for about 5 weeks. One way to reconcile these differences is to compute average treatment effects per week, following (27).

We now write (p.7): “The average school in São Paulo State remained closed for approximately 35 weeks throughout 2020. As such, our estimates imply that students lost approximately 0.009 standard-deviations of learning each week, relative to in-person classes. This effect size is only slightly higher than that in (27) but at least 4-fold that in (21).” We continue: “Estimates in (27) imply a learning loss of 0.3 to 0.4 s.d a year, which translates to 0.005-0.007 s.d a week.” When discussing the effects of reopening schools for in-person classes (pp.8), we write: “In municipalities that authorized schools to reopen for in-person academic activities in 2020, the average school could have done so for at most 5 weeks. Thus, resuming in-person classes contributed to an increase in test scores of about 0.005 s.d a week.”

When it comes to intensity, it is important to bear in mind that our estimate of the treatment effects of school reopening is based on intention-to-treat (ITT) estimates, as we do not have data on which schools actually reopened (and for exactly how long) in the municipalities that issued authorization decrees. As such, we believe this effect size is actually substantially high – in fact, just as large as weekly learning losses from school closures estimated in (27). ”

References

1. G. Lichand, C. Alberto Doria, J. Cossi, O. Leal Neto, Reopening Schools in the Pandemic Did Not Increase COVID-19 Incidence and Mortality in Brazil. *Joao Paulo and Leal Neto, Onicio, Reopening Schools in the Pandemic Did Not Increase COVID-19 Incidence and Mortality in Brazil (March 25, 2021)* (2021).
2. Datafolha, “Educação não presencial na perspectiva dos estudantes e suas famílias, onda 6”, tech. rep. (Fundação Lemann and Itau Social, 2021).
3. J. Angrist, D. Autor, A. Pallais, Marginal Effects of Merit Aid for Low-Income Students. *Working paper* (2021).
4. B. Fitzpatrick, J. Ferrare, M. Berends, J. Waddington, Virtual Illusion: Comparing Student Achievement and Teacher Characteristics in Online and Brick-and-Mortar Charter Schools. *Educational Researcher* **49** (2020).
5. J. Ahn, A. Mceachin, Student Enrollment patterns and Achievement in Ohio’s online charter schools. *Educational Researcher* **46** (2017).
6. R. Zimmer *et al.*, “Charter Schools in Eight States: Effects on Achievement, Attainment , Integration, And Competition”, tech. rep. (Rand Corporation, 2009).
7. C. for Research on Education Outcomes, “Online Charter School Study - 2015”, tech. rep. (Stanford University, 2015).
8. E. Bettinger, L. Fox, S. Loeb, E. Taylor, Virtual Classrooms: How Online College Courses Affect Student Success. *American Economic Review* **7** (2017).
9. N. Bianchi, Y. Lu, H. Song, “The Effect of Computer-Assisted Learning on Students’ Long-Term Development”, tech. rep. No. 28180 (NBER Working Paper, 2020).
10. I. Chirikov, T. Semenova, N. Maloshonok, B. Bettinger, R. Kizilcec, Online education platforms scale college STEM instruction with equivalent learning outcomes at lower cost. *Science Advances* **6** (2020).
11. J. Johnston, C. Ksoll, “Effectiveness of Interactive Satellite-Transmitted Instruction: Experimental Evidence from Ghanaian Primary Schools”, tech. rep. No. 17-08 (CEPA Working Paper, 2017).
12. J. P. Azevedo, A. Hasan, D. Goldemberg, S. A. Iqbal, K. Geven, *Simulating the potential impacts of COVID-19 school closures on schooling and learning outcomes: A set of global estimates* (The World Bank, 2020).
13. *Plataforma de apoio à aprendizagem*, <https://apoioaprendizagem.caeddigital.net/#!/funciona>, Accessed: 2021-05-21.
14. E. A. Hanushek, L. Woessmann, The economic impacts of learning losses. (2020).

15. E. Dorn, B. Hancock, J. Sarakatsannis, E. Viruleg, COVID-19 and learning loss—disparities grow and students need help. *McKinsey & Company, December 8* (2020).
16. J. E. Maldonado, K. De Witte, The effect of school closures on standardised student test outcomes. *KU Leuven Department of Economics Discussion Paper DPS20 17* (2020).
17. C. Boruchowicza, S. Parker, L. Robbins, Time Use of Youth during a Pandemic: Evidence from Mexico. *Working paper* (2021).
18. C. associates, “Understanding Student Needs: Early Results from Fall Assessments”, tech. rep. (Working Paper, 2020).
19. V. Kogan, S. Lavertu, The COVID-19 Pandemic and Student Achievement on Ohio’s Third-Grade English Language Arts Assessment. *Working paper* (2020).
20. B. Domingue, H. Hough, D. Lang, J. Yeatman, Changing Patterns of Growth in Oral Reading Fluency During the COVID19 Pandemic. *Working paper* (2020).
21. M. Kuhfeld, B. Tarasawa, A. Johnson, K. Lewis, Learning during COVID-19: Initial findings on students’ reading and math achievement and growth. *Working paper* (2020).
22. M. Kuhfeld *et al.*, Projecting the potential impact of COVID-19 school closures on academic achievement. *Educational Researcher* **49**, 549–565 (2020).
23. S. J., N. Mahler, B. Fauth, M. Lindner, Did Students Learn Less During the COVID-19 Pandemic? Reading and Mathematics Competencies Before and After the First Pandemic Wave. *Working paper* (2021).
24. T. M., L. Helbling, U. Moser, Educational gains of in-person vs. distance learning in primary and secondary schools: A natural experiment during the COVID-19 pandemic school closures in Switzerland. *International journal of psychology* **56** (2020).
25. M. Percoco, Health shocks and human capital accumulation: the case of Spanish flu in Italian regions. *Regional Studies* **50**, 1496–1508 (2016).
26. V. Amorim, C. Piza, I. J. Lautharte Junior, The Effect of the H1N1 Pandemic on Learning. (2020).
27. P. Engzell, A. Frey, M. D. Verhagen, Learning loss due to school closures during the COVID-19 pandemic. *Proceedings of the National Academy of Sciences* **118** (2021).
28. P. Gouédard, B. Pont, R. Viennet, Education responses to COVID-19: Implementing a way forward. (2020).
29. P. M.S., D. Nogueira, Análise da matriz de competência em uma aplicação real da Avaliação da Aprendizagem em Processo de Língua Portuguesa da Secretaria da Educação do Estado de São Paulo. *Revista de Estudos Linguísticos* (2017).

30. P. M.S., D. Nogueira, *Analisando a ferramenta Avaliação de Aprendizagem em Processo da Secretaria da Educação Básica do Estado de São Paulo como método de ensino de língua materna. Revista de Estudos Linguísticos* (2017).
31. *Mapas de Foco da BNCC*, <https://institutoreuna.org.br/projeto/mapas-de-foco-bncc>, Accessed: 2021-05-21.
32. G. Lichand, J. Christen, *Behavioral Nudges Prevent Student Dropouts in the Pandemic*, 2021.

Decision Letter, first revision:

2nd December 2021

Dear Dr Lichand,

Thank you once again for your manuscript, entitled "The Impacts of Remote Learning in Secondary Education during the Pandemic in Brazil," and for your patience during the peer review process.

Your manuscript has now been evaluated by 3 reviewers, whose comments are included at the end of this letter. Although two reviewers now recommend publication of your work, one reviewer has several remaining concerns. We remain interested in the possibility of publishing your study in *Nature Human Behaviour*, but would like to consider your response to these concerns in the form of a revised manuscript before we make a decision on publication.

Specifically, Reviewer #3 is still concerned about the reliability of your dropout proxies. While we do not believe that you should remove these analyses altogether, we ask you to be fully transparent about the limitations of your proxy, and to report full statistics and figures of your supporting analyses in Supplementary Sections A1 and A2. In your revision, we also ask you to strengthen your argument showing that matching has appropriately controlled for selection effects, as requested by Reviewer #3.

In sum, we invite you to revise your manuscript taking into account all reviewer and editor comments. We are committed to providing a fair and constructive peer-review process. Do not hesitate to contact us if there are specific requests from the reviewers that you believe are technically impossible or unlikely to yield a meaningful outcome.

We hope to receive your revised manuscript within four to eight weeks. We understand that the COVID-19 pandemic is causing significant disruption for many of our authors and reviewers. If you cannot send your revised manuscript within this time, please let us know - we will be happy to extend the submission date to enable you to complete your work on the revision.

- Include a "Response to the editors and reviewers" document detailing, point-by-point, how you addressed each editor and referee comment. If no action was taken to address a point, you must provide a compelling argument. This response will be used by the editors to evaluate your revision and sent back to the reviewers along with the revised manuscript.
- Highlight all changes made to your manuscript or provide us with a version that tracks changes.

[REDACTED]

We look forward to seeing the revised manuscript and thank you for the opportunity to review your work. Please do not hesitate to contact me if you have any questions or would like to discuss these revisions further.

Sincerely,

Arunas Radzvilavicius, PhD
Editor
Nature Human Behaviour

Reviewer expertise:

Reviewer #1:

Reviewer #2:

Reviewer #3:

REVIEWER COMMENTS:

Reviewer #1:

Remarks to the Author:

The authors have addressed my comments and I am happy to recommend publication.

Reviewer #2:

Remarks to the Author:

Dear authors

Thank you very much for revised version. Although I was skeptical in my first review, your reply and changes have significantly changed my mind. You did an excellent job in the revision. I fully agree with your reply to my questions and the implemented changes in the paper. I also liked the multiple additional robustness tests and the more detailed sub-group analyses. I have two remaining, though minor, issues. Nevertheless, I think that implementing them might increase the attractiveness (and hence, citations) of the paper even further.

Reading the comments and replies to the other referees, I agree with reviewer 1 that you embedded the paper better in the literature. With respect to this, there are recently a few new papers published on the theme that you might want to include (e.g. Donnelly and Patrinos; Werner and Woessmann; Grewenig, Lergetporer, Werner, Zierow and Woessmann; Gambi and De Witte; Iterbeke and De Witte – And references in these recent articles). Adding these will embed the paper also in the most recent literature on the covid related learning losses. Moreover, some papers are recently published and do not appear as working papers any longer. Please check in the reference list.

In the revised version, you discuss on page 6 the selection effect. I think that you underplay this effect here. At the very least, you should refer to the recent paper by Werner and Woessmann (the legacy of covid-19 in education) who devote significant attention to the cohort effects and how this underestimates the true learning losses.

Reviewer #3:

Remarks to the Author:

1. "Defining students to be at risk of dropout if they do not take a quarterly exam applied online during the pandemic strikes me as not very convincing indicator of dropout risk."
Figure S.A.1 shows the correlation between actual dropouts and the proposed proxy using taking a quarterly exam in the 4th quarter of 2019. As can be seen in the graph, there is some correlation but hardly a very high one. Showing some correlation (0.7 is not that high actually) is not sufficient to

argue that a variable is a good proxy. This graph overall I consider evidence that using taking the exam is a poor proxy for actual dropout.

The authors provide a section in the supplementary materials on page 2 (section A2) describing how they test that not taking the exam can proxy dropout during the pandemic. Here they do not show a correlation or graph but construct an alternative proxy to dropout measured not as dropout but as the probability of not attending class because in fact they do not have information on actual dropout.

It is difficult to understand why the authors insist on studying the effects of the pandemic on dropout when they do not have information on dropout. The proxies reflect jointly current attendance, knowledge about when the test would be applied/whether test perceived as important, incidence of health problems which affect getting to school etc. etc. I suggest dropping this analysis all together.

2. There really needs to be some basic description of the characteristics of those who take the test and those who don't before and after in 2019 and 2020 to provide an initial idea if selection changes over time and during the pandemic and whether this selection affects the results of the paper. Such an analysis is a basic ingredient to establishing whether the changes in test scores the paper describes in fact reflect effects of online learning/pandemic or are simply differences in the characteristics of the population taking the tests e.g. selection bias. The paper simply does not provide sufficient evidence that the estimates are (at least largely) free of selection bias. Where is the evidence/arguments that the matching has adequately controlled for selection?

Author Rebuttal, first revision:

Reviewer 2

Dear authors, Thank you very much for revised version. Although I was skeptical in my first review, your reply and changes have significantly changed my mind. You did an excellent job in the revision. I fully agree with your reply to my questions and the implemented changes in the paper. I also liked the multiple additional robustness tests and the more detailed sub-group analyses. I have two remaining, though minor, issues. Nevertheless, I think that implementing them might increase the attractiveness (and hence, citations) of the paper even further.

Reading the comments and replies to the other referees, I agree with reviewer 1 that you embedded the paper better in the literature. With respect to this, there are recently a few new papers published on the theme that you might want to include (e.g. Donnelly and Patrinos; Werner and Woessmann; Grewenig, Lergetporer, Werner, Zierow and Woessmann; Gambi and De Witte; Iterbeke and De Witte – And references in these recent articles). Adding these will embed the paper also in the most recent literature on the covid related learning losses. Moreover, some papers are recently published and do not appear as working papers any longer. Please check in the reference list.

Thank you for your invaluable comments, which have helped us greatly improve the paper in the revision process. In the new version of the manuscript, we have updated the introduction to incorporate the suggested references and additional ones. Now, in page 3, we write:

“Quantifying learning losses due to remote learning within primary and secondary education is urgent, as governments need to make informed decisions when trading off the potential health risks of reopening schools in the pandemic (1) against its potential educational benefits. This remains to be the case even with high immunization coverage; in Brazil, while 49.4% of the population had received at least the first shot of the COVID-19 vaccine by July 2021, only approximately 25% of students had returned to in-person classes at that time (2). Several papers attempt to quantify learning losses from remote relative to in-person classes before the pandemic, but with important limitations when it comes to generalizability. Most studies are based on developed country settings (3–5). Out of those, some contrast online to in-person instruction within tertiary education (6–11), while those that focus on secondary schools restrict attention to charter schools, contrasting online to in-person instruction within a very selected sets of students (12–15). In contrast, evidence for developing countries is thinner, and mostly from experiments that use remote learning to expand educational access to rural and remote regions that had no access to education before (16–19) – a very different counterfactual than in-person classes before the pandemic. In turn, the studies that try to estimate the extent of learning losses due to remote learning during the pandemic either rely on simulations and structural models (20–22) or suffer from comparability issues – contrasting different tests and student populations before and during the pandemic, and without parsing out other direct effects of COVID-19, above and beyond the transition to remote learning (23–33). Even the few studies that

rely on appropriate counterfactuals to study this question have to rest on strong assumptions, given the nature of the variation they use to identify causal effects; in particular, differences in the length of school recess across geographical units or that induced by previous epidemics (34, 35) are only loosely related to the changes in instruction mode observed in the context of COVID-19. As such, the only credible available evidence for the impacts of remote learning on secondary schools during the pandemic is for developed countries (36, 37) – leaving key questions unanswered, from the extent of its impacts on student dropouts (an issue relatively unimportant in developed countries, but critical in the developing world) to the extent of heterogeneity of those impacts with respect to student characteristics, such as age, gender and race.”

In the revised version, you discuss on page 6 the selection effect. I think that you underplay this effect here. At the very least, you should refer to the recent paper by Werner and Woessmann (the legacy of covid-19 in education) who devote significant attention to the cohort effects and how this underestimates the true learning losses.

Thank you for flagging this. We now discuss the selection effect more thoroughly in the main text. We also added new analyses to document the nature of selection and how our matching procedures adequately address it, and estimate the sensitivity of our findings to selection using a balanced panel.

In page 7, we now write: “(22) documents, in a different context, that differences between cohorts over time, in particular due to selection in the context of COVID-19, can generate sizable differences in measured learning outcomes throughout the pandemic. Our empirical analysis not only compares how the same cohorts evolved over time, but also, our matching strategy in Columns (4-5) ensures that the characteristics of students who took different exams are balanced over time (see Table S.E.1 in the Supplementary Materials). Results are very robust to the matching procedure. This is not because selection is unimportant – Table S.E.2 in the Supplementary Materials shows that student characteristics matter significantly for differential take-up of standardized tests in 2020 –; rather, this presumably largely reflects the fact that the nature of selection does not change across Q1 and Q2-Q4/2020. As such, self-selection into the exams has small to no impacts on our main results. Table S.E.7 additionally re-estimates the results in Table 1 using a balanced panel, by restricting attention to the students who took *all* standardized tests in 2019 and 2020. Even within this highly selected sub-sample (given the results in Panel A of Table S.E.7), the effect size of remote learning on learning outcomes is still over 70% that within the whole sample, corroborating that findings are not an artifact of selection in the context of our study.”

In Section E of the Supplementary Materials, Table S.E.1 discusses imbalances without matching, and how our matching procedures adequately address them. We write: “Columns (1-2) in Table S.E.1 report mean values for student characteristics, separately for those who took any standardized test in 2019 (column 1) and in 2020 (column 2). While most differences seem small, one can see that, in 2020, not only attendance and grades were higher among the sample with standardized test scores, but also, their characteristics indicate that they

are indeed positively selected: there is a higher share of white students among test-takers, who also tend to come from higher-income schools and more likely to have offered online activities prior to the pandemic. Differences across the two samples are indeed highly statistically significant (p-value of an F-test of joint significance = 0.000). Next, columns (3-4) report characteristics of the sample of 2020 after applying matching procedures: column (3) displays those using inverse probability weights, and column (4), controlling for the propensity score. Sample means under both procedures approximate those of the 2019 sample to a much better extent; in particular, one can see that the proportion of white students, the average per capita income of the school neighborhood and the share of test-takers from schools with online activities prior to the pandemic are (nearly) identical across matched samples. As a result, we no longer reject the hypothesis that the samples are balanced at conventional significance levels in each case (p=0.52 and 0.31, respectively).”

Table S.E.1: Student and school characteristics among those who took standardized tests in 2019 and 2020, with and without matching

	2019 sample	2020 sample	2020 sample with inverse probability weighting	2020 sample controlling for propensity scores
Attendance	0.89	0.95	0.91	0.91
Scorecard grades	6.21	6.39	6.20	6.21
Male	0.48	0.51	0.51	0.51
White	0.55	0.59	0.55	0.56
Per capita income (R\$)	913.00	920.38	912.93	913.08
Prior online activities	0.72	0.74	0.72	0.72
p-value(F-test)		0.00	0.52	0.31

Also in Section E of the Supplementary Materials, Table S.E.7 analyzes the extent to which selection might affect our estimates of the impacts of remote learning on standardized test scores. We write: “Table S.E.7 reports sensitivity tests for selection effects. For this analyses, we track students who took standardized tests in Q4/2019 and in Q4/2020. In Panel A, we estimate how standardized test scores differ for those students, relative to other students, to document the extent of selection. In Panel B, we re-estimate treatment effects of remote learning on standardized test scores using differences-in-differences with a balanced panel, restricting attention to students who took all exams.”

“Panel A shows that standardized test scores in 2019 were 0.09 standard-deviations higher in the selected sample relative to other students (significant at the 1% level), confirming that they are indeed positively selected. In turn, Panel B shows that, even among this highly selected sample, the effects of remote learning were substantially negative. We document that learning losses relative to in-person classes were of the order of 0.225 s.d. (significant at the 1% level) – over 70% of the coefficient reported in Table 1.”

Table S.E.7: Sensitivity tests for selection effects

Panel A: Standardized scores in 2019		
Took the test in 2020	0.092*** (0.0002)	0.089*** (0.0002)
N	2,232,676	
Panel B: Standardized scores in 2020		
Remote learning	-0.222*** (0.0002)	-0.225*** (0.0002)
N	6,142,212	
Grade fixed-effects	no	yes

Reviewer 3

1. “Defining students to be at risk of dropout if they do not take a quarterly exam applied online during the pandemic strikes me as not very convincing indicator of dropout risk.” Figure S.A.1 shows the correlation between actual dropouts and the proposed proxy using taking a quarterly exam in the 4th quarter of 2019. As can be seen in the graph, there is some correlation but hardly a very high one. Showing some correlation (0.7 is not that high actually) is not sufficient to argue that a variable is a good proxy. This graph overall I consider evidence that using taking the exam is a poor proxy for actual dropout.

The authors provide a section in the supplementary materials on page 2 (section A2) describing how they test that not taking the exam can proxy dropout during the pandemic. Here they do not show a correlation or graph but construct an alternative proxy to dropout measured not as dropout but as the probability of not attending class because in fact they do not have information on actual dropout.

It is difficult to understand why the authors insist on studying the effects of the pandemic on dropout when they do not have information on dropout. The proxies reflect jointly current attendance, knowledge about when the test would be applied/whether test perceived as important, incidence of health problems which affect getting to school etc. etc. I suggest dropping this analysis all together.

Thank you for your concerns with this critical issue. While it would have been simpler to just drop the analyses, we were motivated to strengthen the arguments for the use of the proxy due to the complete absence of direct evidence on the effects of remote learning on student dropouts during the pandemic, particularly in developing countries. This is a critical gap in the public debate and policy discussions, one which continues to influence decisions to keep schools open or not as the world is hard hit by new variants.

In response to your comments, we have now expanded our analyses of this proxy extensively in Section A of the Supplementary Materials. In these analyses, we added new administrative data from Q1/2021 that supports our claims about the predictive power of the proxy even during the pandemic. We also corrected our estimates for measurement error directly, using analytical formulas that consider how the effects of classification error on our estimates can be parsed out by computing false positive and false negatives based on administrative outcomes, both prior to and during the pandemic. Last, we also tried to appropriately caveat for the use of the proxy in the Discussion section, pointing out the limitations from not observing actual dropouts in the context of our study.

Let us expand on each of the points above. In page 6, we now write: “In Section A of Supplementary Materials we show that this finding is robust to correcting for measurement error based on administrative data – which allows us to compute false positives and false negatives both prior to and during the pandemic –, and provide evidence that the proxy is indeed highly predictive of not attending any classes in Q1/2021, when in-person classes had been authorized

to return by all municipalities of São Paulo State.”

We added two new subsections to Section A. Section A.2 computes odds ratios comparing the probability of participation in school activities of students who had missing scorecard grades in the previous year to that of other students. This subsection does that for actual re-enrollment in 2020, to validate the proxy with pre-pandemic data, as well as for zero attendance and missing scorecard grades in Q1/2021, to validate it in the context of COVID-19. In that subsection, we write: “Table S.A.1 reports the results. Column (1) shows that students with missing math and Portuguese scorecard grades in 2019 are seven times more likely not to be enrolled in the following year than other students, validating the proxy prior to the pandemic. Since the connection between our proxy and actual dropouts might have changed during such exceptional times – concretely, many students might have failed to hand in homework and take exams during the pandemic due to atypical circumstances, from connectivity constraints to concerns about being exposed to COVID-19 – columns (2) and (3) estimate odds ratios for indicators of school participation in 2021, when in-person classes had been authorized to return by all municipalities of São Paulo State. We focus on whether students engaged in *any* academic activity during this school quarter: attending classes or taking exams that would qualify for scorecard grades in Q1/2021, across all school subjects. If a student missed all classes across all subjects during the whole school quarter, this is essentially equivalent to not being enrolled in school. If our proxy still predicts intended non-enrollment in 2020, we would expect that students at high dropout risk during that year are less likely to participate in any academic activity in 2021. The table shows that students with missing scorecard grades in Q4/2020 were 8.6 times more likely not to have attended a single class (column 2) and 9.7 times more likely not to have taken a single exam (column 3) in Q1/2021. We reject the null hypothesis that each odds ratio is equal to one at the 1% level. As such, we conclude that this proxy remains a strong predictor of student dropouts even throughout the pandemic.”

Table S.A.1: Odds ratios for different measures of student engagement relative to missing scorecard grades

	(1)	(2)	(3)
	Not enrollment in 2020	No classes attended in Q1 2021	No tests taken in Q1 2021
Odds Ratio	7.21*** (0.001)	8.65*** (0.001)	9.78*** (0.001)
N	1,969,552	1,606,909	

Next, subsection A.3 assesses the sensitivity of our results to directly correcting for measurement error from not directly observing actual dropouts. It does so by deriving an analytical formula for the effects of classification error on

the coefficients we estimate. This formula depends on the false positive and false negative rates from using our proxy. While we, of course, cannot compute these rates based on actual dropouts in 2021 (otherwise we would not need a proxy after all), we can compute them in 2020, and approximate them in Q1/2021 using other measures of school participation – concretely, whether students had *zero attendance across all classes* in the school quarter, and whether they had *missing scorecard grades for all classes* in that quarter. We write: “Table S.A.2 reports estimates for treatment effects on proxy dropouts corrected for measurement error according to the procedure above. Column (1) assumes that measurement error is constant across X , and corrects coefficients using equation (10). Column (2) allows measurement error to be correlated with covariates – in particular, it allows it to vary by grade or school quarter –, correcting coefficients using equation (9). Panel A estimates false positive and false negative rates based on actual dropouts in 2020; Panel B, based on zero attendance in Q1 2021; and Panel C, based on no scorecard grades in Q1/2021. All corrected estimates are very close to the ones estimated in Table 1; if anything, corrected estimates are 4-12% higher than coefficients reported in Table 1.”

Table S.A.2: Effects of remote learning on dropout rates correcting for measurement error

	(Q4 2020-Q1 2020)-(Q4 2019-Q1 2019) (1)	(2)
Panel A: Actual Dropouts in 2020		
Remote learning	0.070*** (0.0002)	0.068*** (0.0002)
Panel B: Zero attendance in Q1/2021		
Remote learning	0.068*** (0.0002)	0.067*** (0.0002)
Panel C: Missing all scorecard grades in Q1/2021		
Remote learning	0.065*** (0.0002)	0.065*** (0.0002)
N	8,543,856	
Grade fixed-effects	yes	yes
Matching	no	no

Last, we introduced a paragraph on the limitations of relying on that proxy in the Discussion section. In page 9, we now write: “A limitation of our analyses is that we will not know the extent of actual dropouts until later in 2022 (or even only in 2023), when school systems will no longer re-enroll students automatically (which happened exceptionally in the context of the pandemic). Alternatively, we have relied on missing scorecard grades, a proxy used by the Education Secretariat and its philanthropic partners to identify students at high dropout risk. This proxy is also in line with the literature that attempts to predict student dropouts by measuring their lack of engagement with school activities (38). Section A of Supplementary Materials documents that this proxy is, in fact, predictive of actual dropouts. In 2019, students with missing scorecard grades were approximately seven times more likely not to be enrolled in

the following year than other students. Nevertheless, the proxy is not flawless. Relying on it generates both false positives and false negatives even during in-person classes. Moreover, during the pandemic, missing scorecard grades might rather reflect transitory shocks that prevent those students from handing in homework or taking exams but not necessarily imply that they will drop out of school. Having said that, using administrative data for 2021's first school quarter – when in-person classes had been authorized to return by all municipalities of São Paulo State –, Section A of Supplementary Materials shows that students with missing scorecard grades in the previous year were almost nine times more likely not to attend a single class across all subjects in the following year, relative to other students – corroborating the validity of our proxy even during the pandemic. It also estimates the effects of remote learning correcting directly for classification error, leveraging administrative data that allows computing false positive and false negative rates both prior to and during the pandemic. In any case, using this proxy introduces an additional layer of uncertainty in the estimates, and makes it difficult to directly compare our results with other estimates in the literature based on student dropouts measured from actual re-enrollment decisions. Most importantly, as emphasized by (22), the long-term prospects of these students might still be altered by targeted public policies, from remedial education to cash transfers to active search for out-of-school children and adolescents.”

2. There really needs to be some basic description of the characteristics of those who take the test and those who don't before and after in 2019 and 2020 to provide an initial idea if selection changes over time and during the pandemic and whether this selection affects the results of the paper. Such an analysis is a basic ingredient to establishing whether the changes in test scores the paper describes in fact reflect effects of online learning/pandemic or are simply differences in the characteristics of the population taking the tests e.g. selection bias. The paper simply does not provide sufficient evidence that the estimates are (at least largely) free of selection bias. Where is the evidence/arguments that the matching has adequately controlled for selection?

We thank you for raising that concern. The previous version of the paper already contained Table S.E.1, which provides descriptive evidence about differences in characteristics of the students who took standardized tests in 2020 relative to those who took them in 2019. We apologize for not giving it enough emphasis in the previous version of the manuscript.

Table S.E.1: Selection into Q4 standardized test (across all grades)

	Marginal probability change
White x 2020	0.051*** (0.001)
Male x 2020	0.005*** (0.001)
Scorecard grade x 2020 (10 scale)	0.045*** (0.001)
Scorecard frequency x 2020 (100 scale)	-0.001 (0.001)
Income x 2020 (thousand R\$)	0.011*** (0.001)

In response to your comment, we have now added new analyses to document the nature of selection and how our matching procedures adequately address it, and estimate the sensitivity of our findings to selection using a balanced panel. We also discuss the selection effect more thoroughly in the main text. In page 7, we now write: “(22) documents, in a different context, that differences between cohorts over time, in particular due to selection in the context of COVID-19, can generate sizable differences in measured learning outcomes throughout the pandemic. Our empirical analysis not only compares how the same cohorts evolved over time, but also, our matching strategy in Columns (4-5) ensures that the characteristics of students who took different exams are balanced over time (see Table S.E.1 in the Supplementary Materials). Results are very robust to the matching procedure. This is not because selection is unimportant – Table S.E.2 in the Supplementary Materials shows that student characteristics matter significantly for differential take-up of standardized tests in 2020 –; rather, this presumably largely reflects the fact that the nature of selection does not change

across Q1 and Q2-Q4/2020. As such, self-selection into the exams has small to no impacts on our main results. Table S.E.7 additionally re-estimates the results in Table 1 using a balanced panel, by restricting attention to the students who took *all* standardized tests in 2019 and 2020. Even within this highly selected sub-sample (given the results in Panel A of Table S.E.7), the effect size of remote learning on learning outcomes is still over 70% that within the whole sample, corroborating that findings are not an artifact of selection in the context of our study.”

In Section E of the Supplementary Materials, Table S.E.1 discusses imbalances without matching, and how our matching procedures adequately address them. We write: “Columns (1-2) in Table S.E.1 report mean values for student characteristics, separately for those who took any standardized test in 2019 (column 1) and in 2020 (column 2). While most differences seem small, one can see that, in 2020, not only attendance and grades were higher among the sample with standardized test scores, but also, their characteristics indicate that they are indeed positively selected: there is a higher share of white students among test-takers, who also tend to come from higher-income schools and more likely to have offered online activities prior to the pandemic. Differences across the two samples are indeed highly statistically significant (p-value of an F-test of joint significance = 0.000). Next, columns (3-4) report characteristics of the sample of 2020 after applying matching procedures: column (3) displays those using inverse probability weights, and column (4), controlling for the propensity score. Sample means under both procedures approximate those of the 2019 sample to a much better extent; in particular, one can see that the proportion of white students, the average per capita income of the school neighborhood and the share of test-takers from schools with online activities prior to the pandemic are (nearly) identical across matched samples. As a result, we no longer reject the hypothesis that the samples are balanced at conventional significance levels in each case (p=0.52 and 0.31, respectively).”

Table S.E.1: Student and school characteristics among those who took standardized tests in 2019 and 2020, with and without matching

	2019 sample	2020 sample	2020 sample with inverse probability weighting	2020 sample controlling for propensity scores
Attendance	0.89	0.95	0.91	0.91
Scorecard grades	6.21	6.39	6.20	6.21
Male	0.48	0.51	0.51	0.51
White	0.55	0.59	0.55	0.56
Per capita income (R\$)	913.00	920.38	912.93	913.08
Prior online activities	0.72	0.74	0.72	0.72
p-value(F-test)		0.00	0.52	0.31

Also in Section E of the Supplementary Materials, Table S.E.7 analyzes the

extent to which selection might affect our estimates of the impacts of remote learning on standardized test scores. We write: “Table S.E.7 reports sensitivity tests for selection effects. For this analyses, we track students who took standardized tests in Q4/2019 and in Q4/2020. In Panel A, we estimate how standardized test scores differ for those students, relative to other students, to document the extent of selection. In Panel B, we re-estimate treatment effects of remote learning on standardized test scores using differences-in-differences with a balanced panel, restricting attention to students who took all exams.”

“Panel A shows that standardized test scores in 2019 were 0.09 standard-deviations higher in the selected sample relative to other students (significant at the 1% level), confirming that they are indeed positively selected. In turn, Panel B shows that, even among this highly selected sample, the effects of remote learning were substantially negative. We document that learning losses relative to in-person classes were of the order of 0.225 s.d. (significant at the 1% level) – over 70% of the coefficient reported in Table 1.”

Table S.E.7: Sensitivity tests for selection effects

Panel A: Standardized scores in 2019		
Took the test in 2020	0.092*** (0.0002)	0.089*** (0.0002)
N	2,232,676	
Panel B: Standardized scores in 2020		
Remote learning	-0.222*** (0.0002)	-0.225*** (0.0002)
N	6,142,210	
Grade fixed-effects	no	yes

References

1. G. Lichand, C. Alberto Doria, J. Cossi, O. Leal Neto, Reopening Schools in the Pandemic Did Not Increase COVID-19 Incidence and Mortality in Brazil. *Joao Paulo and Leal Neto, Onicio, Reopening Schools in the Pandemic Did Not Increase COVID-19 Incidence and Mortality in Brazil (March 25, 2021)* (2021).
2. Datafolha, “Educação não presencial na perspectiva dos estudantes e suas famílias, onda 6”, tech. rep. (Fundação Lemann and Itau Social, 2021).
3. E. Grewenig, P. Lergetporer, K. Werner, L. Woessmann, L. Zierow, Covid-19 and Educational Inequality: How School Closures Affect Low- and High-Achieving Students. *CESifo Working Papers* (2020).
4. L. Gambi, K. D. Witte, The resiliency of school outcomes after the COVID-19 pandemic. Standardised test scores and inequality one year after long term school closures. *Working Paper* (2021).
5. K. Iterbeke, K. D. Witte, Helpful or Harmful? The Role of Personality Traits in Student Experiences of the COVID-19 Crisis and School Closure. *Personality and Social Psychology Bulletin* (2021).
6. D. Figlio, M. Rush, L. Yin, Is It Live or Is It Internet? Experimental Estimates of the Effects of Online Instruction on Student Learning. *Journal of Labor Economics* **31** (2013).
7. J. Angrist, D. Autor, A. Pallais, Marginal Effects of Merit Aid for Low-Income Students. *Working paper* (2021).
8. W. T. Alpert, K. A. Couch, O. R. Harmon, A Randomized Assessment of Online Learning. *American Economic Review* **106** (2016).
9. M. P. Cacaault, C. Hildebrand, J. Laurent-Lucchetti, M. Pellizzari, Distance Learning in Higher Education: Evidence from a Randomized Experiment. *Journal of the European Economic Association* **19** (2021).
10. M. S. Kofoed, L. Gebhart, D. Gilmore, R. Moschitto, Zooming to Class?: Experimental Evidence on College Students’ Online Learning during COVID-19. *IZA Working Paper* (2021).
11. K. A. Bird, B. L. Castleman, G. Lohner, Negative Impacts From the Shift to Online Learning During the COVID-19 Crisis: Evidence from a Statewide Community College System. *EDWorking Paper* (2021).
12. B. Fitzpatrick, J. Ferrare, M. Berends, J. Waddington, Virtual Illusion: Comparing Student Achievement and Teacher Characteristics in Online and Brick-and-Mortar Charter Schools. *Educational Researcher* **49** (2020).
13. J. Ahn, A. Mceachin, Student Enrollment patterns and Achievement in Ohio’s online charter schools. *Educational Researcher* **46** (2017).
14. R. Zimmer *et al.*, “Charter Schools in Eight States: Effects on Achievement, Attainment , Integration, And Competition”, tech. rep. (Rand Corporation, 2009).

15. C. for Research on Education Outcomes, “Online Charter School Study - 2015”, tech. rep. (Stanford University, 2015).
16. E. Bettinger, L. Fox, S. Loeb, E. Taylor, Virtual Classrooms: How Online College Courses Affect Student Success. *American Economic Review* **7** (2017).
17. N. Bianchi, Y. Lu, H. Song, “The Effect of Computer-Assisted Learning on Students’ Long-Term Development”, tech. rep. No. 28180 (NBER Working Paper, 2020).
18. I. Chirikov, T. Semenova, N. Maloshonok, B. ger, R. Kizilcec, Online education platforms scale college STEM instruction with equivalent learning outcomes at lower cost. *Science Advances* **6** (2020).
19. J. Johnston, C. Ksoll, “Effectiveness of Interactive Satellite-Transmitted Instruction: Experimental Evidence from Ghanaian Primary Schools”, tech. rep. No. 17-08 (CEPA Working Paper, 2017).
20. J. P. Azevedo, A. Hasan, D. Goldemberg, S. A. Iqbal, K. Geven, *Simulating the potential impacts of COVID-19 school closures on schooling and learning outcomes: A set of global estimates* (The World Bank, 2020).
21. *Plataforma de apoio à aprendizagem*, <https://apoioaprendizagem.caeddigital.net/#!/funciona>, Accessed: 2021-05-21.
22. K. Werner, L. Woessmann, The Legacy of Covid-19 in Education. *CESifo Working Papers* (2021).
23. E. A. Hanushek, L. Woessmann, The economic impacts of learning losses. (2020).
24. E. Dorn, B. Hancock, J. Sarakatsannis, E. Viruleg, COVID-19 and learning loss—disparities grow and students need help. *McKinsey & Company, December* **8** (2020).
25. J. E. Maldonado, K. De Witte, The effect of school closures on standardised student test outcomes. *British Education Research Journal* (2021).
26. C. Boruchowicza, S. Parker, L. Robbins, Time Use of Youth during a Pandemic: Evidence from Mexico. *World Development* (2022).
27. C. associates, “Understanding Student Needs: Early Results from Fall Assessments”, tech. rep. (Working Paper, 2020).
28. V. Kogan, S. Lavertu, The COVID-19 Pandemic and Student Achievement on Ohio’s Third-Grade English Language Arts Assessment. *Working paper* (2020).
29. B. Domingue, H. Hough, D. Lang, J. Yeatman, Changing Patterns of Growth in Oral Reading Fluency During the COVID19 Pandemic. *Working paper* (2020).
30. M. Kuhfeld, B. Tarasawa, A. Johnson, K. Lewis, Learning during COVID-19: Initial findings on students’ reading and math achievement and growth. *Working paper* (2020).

31. M. Kuhfeld *et al.*, Projecting the potential impact of COVID-19 school closures on academic achievement. *Educational Researcher* **49**, 549–565 (2020).
32. S. J., N. Mahler, B. Fauth, M. Lindner, Did Students Learn Less During the COVID-19 Pandemic? Reading and Mathematics Competencies Before and After the First Pandemic Wave. *Working paper* (2021).
33. T. M., L. Helbling, U. Moser, Educational gains of in-person vs. distance learning in primary and secondary schools: A natural experiment during the COVID-19 pandemic school closures in Switzerland. *International journal of psychology* **56** (2020).
34. M. Percoco, Health shocks and human capital accumulation: the case of Spanish flu in Italian regions. *Regional Studies* **50**, 1496–1508 (2016).
35. V. Amorim, C. Piza, I. J. Lautharte Junior, The Effect of the H1N1 Pandemic on Learning. (2020).
36. P. Engzell, A. Frey, M. D. Verhagen, Learning loss due to school closures during the COVID-19 pandemic. *Proceedings of the National Academy of Sciences* **118** (2021).
37. R. Donnelly, H. A. Patrinos, Learning loss during COVID-19: An early systematic review. *Covid Economics* **77** (2021).
38. C. Carter, “Early detection of dropout risk: measuring student engagement at the elementary-school level”, tech. rep. (Graduate Faculty of the University of Georgia, 2007).

Decision Letter, second revision:

Our ref: NATHUMBEHAV-210515479B

22nd February 2022

Dear Dr. Lichand,

Thank you for submitting your revised manuscript "The Impacts of Remote Learning in Secondary Education during the Pandemic in Brazil" (NATHUMBEHAV-210515479B). I am writing to you instead of Dr Radzvilavicius, as he is currently on leave.

Your manuscript has now been evaluated by Reviewer 3 and their comments are below. As you can see, Reviewer 3 finds that the paper has largely improved in revision. We will therefore be happy in principle to publish it in Nature Human Behaviour, pending minor revisions to satisfy Reviewer 3's final requests and to comply with our editorial and formatting guidelines.

We are now performing detailed checks on your paper and will send you a checklist detailing our editorial and formatting requirements within two weeks. Please do not upload the final materials and make any revisions until you receive this additional information from us.

Sincerely,

Samantha Antusch

Samantha Antusch, PhD
Editor
Nature Human Behaviour

Reviewer #3 (Remarks to the Author):

Referee report:

My main critique has focused on the proxy used for dropout risk which during the period of school closure was captured using the probability of taking an online standardized test whereas when schools are open such tests are administered in person. I find the analysis of the proxy on not attending any classes in Q1/2021, when in-person classes had been authorized to return in Sao Paulo State helpful although not completely convincing.

While the pandemic will likely have long term effects on dropout and is an important topic, it is also important that published results are plausibly unbiased, e.g. unlikely to provide misleading estimates not just on whether an effect is positive or negative but also on the size of the magnitudes.

My suggestion is thus to modify the strong tone of the text to say instead of for instance on p. 6 "The table shows that, by all accounts, dropout risk has increased dramatically during remote learning, by roughly 365% (significant at the 1% level; Columns 3-5). "The table is suggestive of important potential effects on dropout, as measured by our proxy indicator for dropout risk." Similarly in the rest of the text.

Minor comment:

Relatedly, please provide some references for the following claim on p. 6 "This proxy has been used for years by the Education Secretary and by philanthropic organizations that support quality education in Brazil to predict student dropouts, especially when it comes to identifying the schools most likely to be affected."

Final Decision Letter:

Dear Dr Lichand,

We are pleased to inform you that your Article "The Impacts of Remote Learning in Secondary Education during the Pandemic in Brazil", has now been accepted for publication in Nature Human Behaviour.

Please note that *Nature Human Behaviour* is a Transformative Journal (TJ). Authors whose manuscript was submitted on or after January 1st, 2021, may publish their research with us through the traditional subscription access route or make their paper immediately open access through payment of an article-processing charge (APC). Authors will not be required to make a final decision about access to their article until it has been accepted. IMPORTANT NOTE: Articles submitted before January 1st, 2021, are not eligible for Open Access publication. Find out more about Transformative Journals

An online order form for reprints of your paper is available at <https://www.nature.com/reprints/author-reprints.html>. All co-authors, authors' institutions and

authors' funding agencies can order reprints using the form appropriate to their geographical region.

With best regards,

Samantha Antusch

Samantha Antusch, PhD
Editor
Nature Human Behaviour